# IL-15 sustains IL-7R-independent ILC2 and ILC3 development

Michelle L. Robinette[1], Jennifer K. Bando[1], Wilbur Song[1], Tyler K. Ulland[1], Susan Gilfillan[1] & Marco Colonna[1]

The signals that maintain tissue-resident innate lymphoid cells (ILC) in different microenvironments are incompletely understood. Here we show that IL-7 receptor (IL-7R) is not strictly required for the development of any ILC subset, as residual cells persist in the small intestinal lamina propria (siLP) of adult and neonatal $Il7ra^{-/-}$ mice. $Il7ra^{-/-}$ ILC2 primarily express an ST2$^-$ phenotype, but are not inflammatory ILC2. CCR6$^+$ ILC3, which express higher Bcl-2 than other ILC3, are the most abundant subset in $Il7ra^{-/-}$ siLP. All ILC subsets are functionally competent *in vitro*, and are sufficient to provide enhanced protection to infection with *C. rodentium*. IL-15 equally sustains wild-type and $Il7ra^{-/-}$ ILC survival *in vitro* and compensates for IL-7R deficiency, as residual ILCs are depleted in mice lacking both molecules. Collectively, these data demonstrate that siLP ILCs are not completely IL-7R dependent, but can persist partially through IL-15 signalling.

[1] Department of Pathology & Immunology, Washington University School of Medicine, 509 S. Euclid Ave Box 8118, St Louis, Missouri 63110, USA. Correspondence and requests for materials should be addressed to M.C. (email: mcolonna@pathology.wustl.edu).

Innate lymphoid cells (ILC) are a lineage of professional cytokine-producing cells that mirror T cells in transcriptional circuitry and effector functions, but derive from distinct progenitors and do not express recombined antigen-specific receptors[1–3]. Primary ILC classes encompass natural killer (NK) cells, which parallel cytotoxic CD8$^+$ T cells, and three additional groups of ILCs enriched at mucosal surfaces that mirror polarized CD4$^+$ helper T cell subsets, called ILC1, ILC2, and ILC3 (refs 1–3). Functionally, Eomes$^-$T-bet$^+$ ILC1, like Eomes$^+$T-bet$^+$ NK cells, produce IFN-γ, whereas GATA3$^+$ ILC2 produce IL-5 and IL-13, and RORγt$^+$ ILC3 produce IL-22 and/or IL-17. In mouse, ILC3 are a particularly diverse lineage, and include fetal lymphoid tissue inducer (LTi) cells, as well as three adult subsets, T-bet$^-$ CCR6$^+$ LTi-like ILC3, T-bet$^{lo}$ CCR6$^-$ NKp46$^-$ (double negative, DN) ILC3, and T-bet$^+$ NKp46$^+$ ILC3 (refs 2,3).

ILC develop downstream of the common lymphoid progenitor (CLP) from a series of progenitors with progressively restricted fates. All ILCs, but not T or B cells, are generated by early innate lymphoid progenitors (EILP)[4] and alpha-lymphoid progenitors (αLP)[5]; common helper ILC precursors (CHILP) generate ILC but not NK cells[6]; and ILC precursors (ILCP) differentiate into ILC1, ILC2, NKp46$^+$ ILC3 and DN ILC3, but not CCR6$^+$ LTi-like ILC3 (ref. 7). In mice, ILC development requires the common cytokine receptor γ-chain (γ_c), which is shared between IL-2, IL-4, IL-7, IL-9, IL-15 and IL-21 (ref. 8). Consensus in the field has been that, as classes of cells, ILC1 and NK cells require IL-15 (refs 9–14), whereas ILC2 and ILC3 rely on IL-7 for development in the bone marrow and/or homeostasis in the periphery. Supporting this notion, ILC2 and ILC3 are greatly reduced in number in IL-7-deficient (Il7$^{-/-}$) or IL-7 receptor α (IL-7Rα) deficient (Il7ra$^{-/-}$) mice[13,15]. Defects are greatest in mice lacking IL-7Rα, which transduces signals for both IL-7 and thymic stromal lymphopoeitin (TSLP)[13,15].

The requirement of IL-7R by the primary ILC subsets defined by lineage studies has not been tested systematically across the mucosal surfaces where most ILC2 and ILC3 are located. It is unknown if microenvironments differentially require IL-7 to support ILC homeostasis, and if so, the identity of the compensatory survival signal. Moreover, the function of residual cells is controversial for ILC3 and unknown for ILC2, the latter of which have not been reported in substantial number in mice lacking IL-7 signalling. Given that ILC2 and ILC3 are widely thought to be IL-7-dependent, these questions are particularly important to establish lineage relationships of new ILC subsets. Here we systematically test the frequency, number, phenotype, and function of ILC2 and ILC3 subsets compared to NK cells and ILC1 in Il7ra$^{-/-}$ mice, which have the most severe defect in ILC generation.

## Results

**siLP sustains residual ILCs in Il7ra$^{-/-}$ mice.** We assessed ILC frequency and total number in Il7ra$^{-/-}$ mice using an intracellular transcription factor (TF) approach (Fig. 1a), first turning our attention to the siLP which contains the greatest number and diversity of ILCs[1,16,17]. Here we detected all ILC subsets in Il7ra$^{-/-}$ mice (Fig. 1a). Compared to wild-type (WT) control, NK cells and ILC1 were significantly increased in frequency among CD45$^+$CD3$^-$CD19$^-$ cells in the lymphocyte gate (Fig. 1b). Meanwhile, ILC2 and NKp46$^+$ ILC3 were significantly reduced in frequency by 3.3- and 3.5-fold, respectively. The frequencies of DN ILC3 and CCR6$^+$ ILC3 subsets were unchanged (Fig. 1b). Consistent with prior reports[6,18–21], the total numbers of all siLP ILC subsets except NK cells and ILC1 were reduced (Fig. 1c). Yet, we noted that the total number of residual siLP Il7ra$^{-/-}$ ILC2 were equal to or greater than that of other ILCP-derived cells, that is, ILC1, DN

ILC3 and NKp46$^+$ ILC3 (Fig. 1c). CHILP-derived CCR6$^+$ ILC3 became the most abundant ILC subset in Il7ra$^{-/-}$ mice (Fig. 1c), in line with their greatest frequency within Il7ra$^{-/-}$ ILCs (Fig. 1b).

We next analysed other ILC-enriched mucosal tissues, namely colon, lung and gonadal adipose tissue. In WT mice, NK cells, ILC1 and ILC2 are major colonic populations, while ILC3 account for only a small fraction of ILCs (Fig. 1d,e). In Il7ra$^{-/-}$ colon, we found a significant increased frequency of NK cells compared to WT control and a decreased frequency of ILC2 by 16.8-fold. DN ILC3 were decreased by 2.1-fold in Il7ra$^{-/-}$ mice. There was no difference in the frequency of ILC1 or NKp46$^+$ and CCR6$^+$ ILC3 (Fig. 1e). All ILCs were reduced in total number in the Il7ra$^{-/-}$ colon. The greatest difference was between ILC2 from WT and Il7ra$^{-/-}$ mice (Fig. 1f). In the lung and gonadal adipose tissue, the frequency of Il7ra$^{-/-}$ ILC2 was reduced by 9.4-fold (Fig. 1g,h) and 52.2-fold (Fig. 1j,k), respectively, resulting in a decrease in the total number of ILC2 compared with WT (Fig. 1i,l). Thus, persistence of substantial numbers of residual ILCs was predominantly a feature of the siLP.

We next asked whether the lower IL-7-dependency we identified in the siLP was based on tissue-specific differences in capacity to utilize IL-7. To test this hypothesis, we assessed the expression of IL-7R (CD127) in siLP, colon, lung and adipose tissue ILCs from individual mice, along with spleen NK cells and ILC1 (Fig. 2a). We found that all ILC1, ILC2, and ILC3 expressed CD127. Shifts in CD127 expression between ILC subsets in different tissues in WT mice (Fig. 2a) did not correlate with changes in ILC frequency or number in Il7ra$^{-/-}$ mice (Fig. 1a–l). In contrast, we noted that the percentage of cells that expressed CD127 was different among NK cells, with colon expressing most, siLP expressing intermediate, and spleen NK cells expressing least amounts of CD127 (Fig. 2a,b). We conclude that tissue NK cells may be more reliant on IL-7R than conventional splenic cells. However, this mechanism does not appear to explain differences in ILC maintenance for ILCs that constitutively express IL-7R.

**ILCs persist in neonatal Il7ra$^{-/-}$ mice.** Immunodeficiency causes immune deregulation and may differently affect the microbial communities that colonize mucosal surfaces, possibly impacting ILC generation and/or homeostasis[22,23]. To determine whether persistent ILC generation in the siLP of Il7ra$^{-/-}$ mice was a genetic or reactive process, we analysed ILC in Il7ra$^{-/-}$ and WT siLP and colon LP on the day of birth, a time when all ILC can easily be detected and microbial colonization is minimal[24,25]. All ILC subsets were present in neonatal Il7ra$^{-/-}$ siLP (Fig. 3a), as in the adult. We appreciated that the majority of WT ILC were ILC3, particularly CCR6$^+$ ILC3 and DN ILC3, consistent with the literature[24,26,27] (Fig. 3a). In comparison to WT neonates, the frequencies of Il7ra$^{-/-}$ ILC2 and each ILC3 subset were decreased, while ILC1 frequency was increased and NK cell frequency was unchanged (Fig. 3a,b). This corresponded to reduced total numbers of ILC2 and ILC3, but unchanged NK cells and ILC1 (Fig. 3c). We conclude that IL-7R deficiency leads to limited ILC generation in both neonatal and adult siLP.

All ILC subsets were also present in Il7ra$^{-/-}$ neonatal colon LP (Fig. 3d). Unlike the limited number of ILC3 in WT adult colon, neonatal colon LP mostly contained ILC3, with DN ILC3 predominating (Fig. 3d). Compared to WT neonates, the frequencies of NK cells, ILC2, DN ILC3 and NKp46$^+$ ILC3 in Il7ra$^{-/-}$ neonates were decreased, while CCR6$^+$ ILC3 frequency was increased and ILC1 was unchanged (Fig. 3e). ILC2, DN ILC3 and NKp46$^+$ ILC3 were also decreased in total number

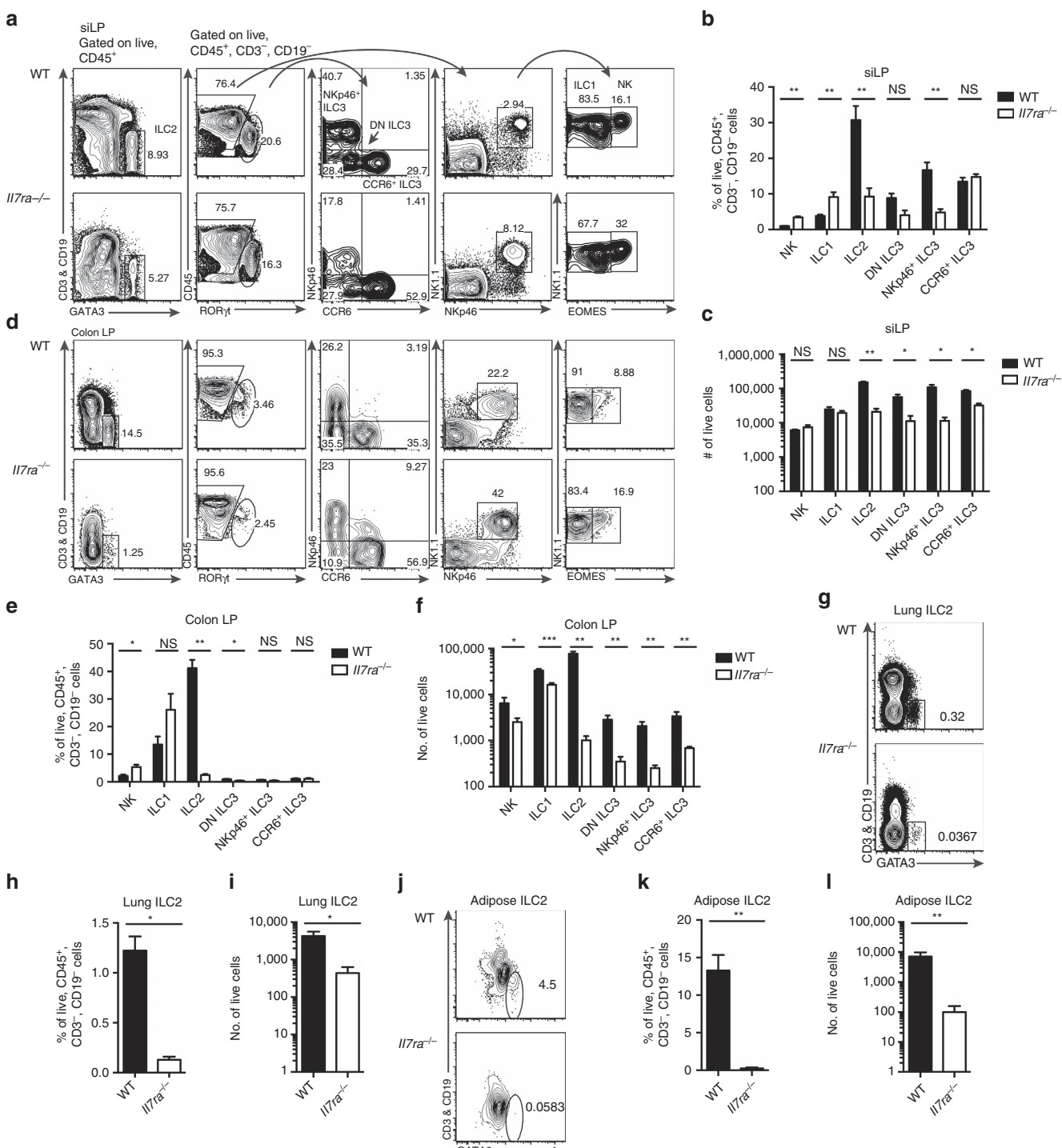

**Figure 1 | siLP sustains residual ILCs in *Il7ra*$^{-/-}$ mice.** (**a–c**) siLP, (**d–f**) colon LP, (**g–i**) lung and (**j–l**) gonadal adipose tissue lymphocytes were isolated from *Il7ra*$^{-/-}$ or WT control mice and indicated ILC subsets assessed. (**a**) Electronic gating strategy and representative flow plots of siLP. (**b**) Frequency and (**c**) total number of ILC subsets from siLP. (**d**) Representative flow plots of colon LP as gated in **a**. (**e**) Frequency and (**f**) total number of ILC subsets from colon LP. (**g–i**) Representative flow plots, (**e**) frequency, and (**f**) total number of ILC2 from lung. (**j**) Representative flow plots, (**k**) frequency, and (**l**) total number of ILC2 from gonadal adipose tissue. Data represent (**a–f,j–l**) $n = 6$ mice per genotype from three independent experiments or (**g–i**) $n = 4$ mice per genotype from 2 independent experiments. Error bars mean ± s.e.m. *$P < 0.05$, **$P < 0.01$, two-tailed Mann–Whitney test.

in *Il7ra*$^{-/-}$ mice, while NK cells, ILC1 and CCR6$^+$ ILC3 were not significantly different (Fig. 3f). Thus, IL-7R is not completely required for neonatal ILC2 or ILC3 generation. Collectively, these data suggest that IL-7R-independent ILC generation is unlikely to be purely reactive to the changes in microbiota or the tissue microenvironment.

**Partially functionally ST2$^-$ ILC2 are not inflammatory ILC2.** ILC2 are often identified among lineage negative cells by cell-surface markers rather than intracellular TFs. As ILC2 have not been reported in *Il7*$^{-/-}$ or *Il7ra*$^{-/-}$ mice[20,21,27–30], we wondered if siLP ILC2 from *Il7ra*$^{-/-}$ mice had an altered cell-surface phenotype. As previously reported[20] we noted three

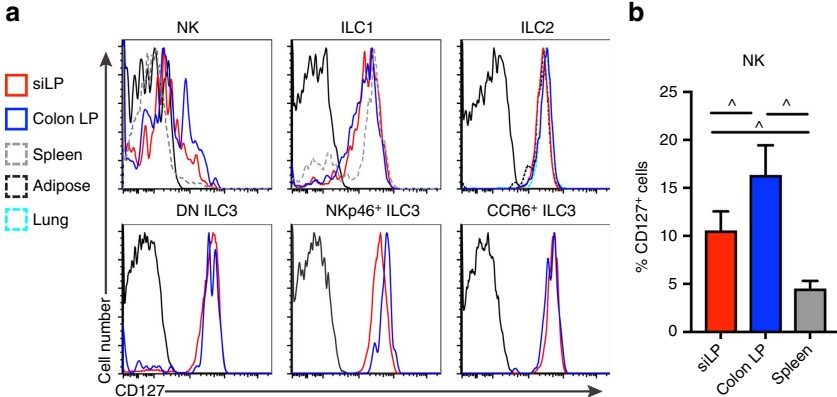

**Figure 2 | CD127 expression by ILC.** (**a,b**) WT CD127 expression was measured across the ILC subsets in Fig. 1a–i, in addition to splenic NKp46[+]NK1.1[+]Eomes[+] NK cells and NKp46[+]NK1.1[+]Eomes[−] ILC1, using biotinylated anti-CD127 with SAv-PeCy7 compared to FMO controls. (**a**) Representative histograms of CD127 expression by ILCs from different organs, with traces corresponding to the legend to the left of the graphs. (**b**) Quantification of CD127 expression by siLP, colon LP, and spleen NK cells between individual mice based on FMO negative control electronic gates. Error bars mean ± s.e.m. Data represent (**a,b**) $n = 9$ mice from three independent experiments. ^$P < 0.05$, RM one-way analysis of variance with Geisser-Greenhouse correction and Tukey's test for multiple comparisons.

populations within siLP GATA3[hi] ILC2 in WT and $Il7ra^{−/−}$ mice: ST2[+]KLRG1[+] ILC2, ST2[−]KLRG1[+] ILC2, and ST2[−]KLRG1[−] ILC2 (Fig. 4a). ST2[+] KLRG1[+] ILC2 were significantly reduced in frequency in $Il7ra^{−/−}$ mice, while ST2[−]KLRG1[−] ILC2 were significantly increased (Fig. 4b). In both genotypes, the majority of cells were ST2[−]KLRG1[+] ILC2, which had equal frequency (Fig. 4b). Neonatal $Il7ra^{−/−}$ GATA3[hi] ILC2 also expressed less ST2 than WT mice (Fig. 4a,c); these cells do not express KLRG1 (ref. 20). Therefore, IL-7R deficiency altered adult and neonatal ILC2 phenotypes.

We next asked whether these subsets were truly ILC2. ILC2 constitutively produce IL-5, which along with IL-13, are enhanced by TSLP, IL-33 and tuft-cell produced IL-25 (refs 28,31–34) that bind the respective receptors IL-7R/TSLPR, ST2 and IL25R. Previously, we evaluated IL-7R and ST2 expression across GATA3[hi] cells (Figs 2a and 4a). All ILC2 subsets expressed cell-surface IL-25R while negative control ILC3 did not (Fig. 4d); IL-25R expression was not different between subsets or genotypes (Fig. 4d). IL-25 has previously been shown to induce ST2[−]KLRG1[+] 'inflammatory ILC2' (iILC2), which persist in $Il7ra^{−/−}$ mice and express intermediate levels of RORγt compared to siLP ILC3 (ref. 35). We measured expression of RORγt in ILC2 subsets compared to positive control CCR6[+] ILC3 and negative control ST2[+]KLRG1[+] ILC2 and found that no ILC2 subset expressed RORγt in WT or $Il7ra^{−/−}$ mice (Fig. 4e). We conclude that siLP ILC2 include ST2[+] and ST2[−] subsets, and have fewer ST2[+] cells in the absence of IL-7R signalling.

The function of WT ST2[−] ILC2 compared to ST2[+] ILC2 as well as that of residual ILC2 in $Il7ra^{−/−}$ mice has not been tested. WT ST2[−] ILC2 and $Il7ra^{−/−}$ ILC2 were unlikely to respond to IL-33 and $Il7ra^{−/−}$ ILC2 were incapable to respond to TSLP, given absent expression of receptors for these cytokines. To compare basal and maximal cytokine production between WT and $Il7ra^{−/−}$ ILC2 subsets, we cultured cells with medium, PMA + ionomycin, IL-25 alone, or IL-25 + IL-33 + TSLP, and measured the frequency of cells that produced IL-5 and IL-13 (Fig. 4f–i). KLRG1[−] ILC2 have previously been shown to be immature cells that do not produce IL-5 or IL-13 in WT mice[20], therefore, we analysed ST2[+]KLRG1[+] and ST2[−]KLRG1[+] subsets. PMA and ionomycin generated the strongest cytokine production from all ILC2 (Fig. 4f–i). The ILC2 subset that

produced the most cytokines across all conditions was WT ST2[+] KLRG1[+] ILC2; $Il7ra^{−/−}$ ST2[+]KLRG1[+] ILC2 and WT ST2[−] KLRG1[+] ILC2 were indistinguishable from each other (Fig. 4f–i). Only WT ST2[+] KLRG1[+] ILC2 were able to substantially produce IL-13 (Fig. 4f,g). $Il7ra^{−/−}$ ST2[−]KLRG1[+] ILC2 produced IL-5 and IL-13 at the lowest frequency (Fig. 4f–i). As a greater frequency of ST2[+] KLRG1[+] ILC2 produced IL-5 and IL-13 compared to ST2[−] KLRG1[+] ILC2 from both WT and $Il7ra^{−/−}$ mice even when IL-33 was not included in the stimulation, we conclude that conditioning of these cells occurs *in vivo*. $Il7ra^{−/−}$ ILC2 were less conditioned than WT ILC2, correlating with the expression of ST2.

**$Il7ra^{−/−}$ ILC3 partially protect against *C. rodentium*.** To test the function of ILC3, we cultured cells with IL-23 and IL-1β *in vitro*, and measured IL-22 and IL-17a production. We found that there was no significant difference in IL-22 production between any ILC3 subset in $Il7ra^{−/−}$ and WT mice (Fig. 5a,b). In comparison, significantly fewer DN and CCR6[+] ILC3 produced IL-17a (Fig. 5c,d). NKp46[+] ILC3 in WT mice produce little IL-17a[26] and there was no significant difference between genotypes (Fig. 5c,d).

Because IL-22 production was equally intact in $Il7ra^{−/−}$ ILC3 *in vitro* (Fig. 5a,b), we wondered if these cells could protect mice from *C. rodentium* given that host defense against this pathogen depends on IL-22 (ref. 36). To perform this experiment, we first generated $Rag1^{−/−}Il7ra^{−/−}$ mice, as $Il7ra^{−/−}$ mice develop some T cells[37] that may compete with ILCs for trophic factors[38]. We infected cohoused $Rag1^{−/−}$, $Rag1^{−/−}Il7ra^{−/−}$ and $Rag2^{−/−}Il2rg^{−/−}$ with a lethal dose of *C. rodentium* and measured weight loss and survival (Fig. 5e,f). From day 4 post-infection through day 6, $Rag2^{−/−}Il2rg^{−/−}$ lost significantly more weight than either $Rag1^{−/−}Il7ra^{−/−}$ or $Rag1^{−/−}$ mice; there was never a difference between weight loss in $Rag1^{−/−}Il7ra^{−/−}$ and $Rag1^{−/−}$ mice (Fig. 5e). This corresponded to a highly significant difference in survival between ILC3-sufficient $Rag1^{−/−}$ and ILC3-depleted $Rag2^{−/−}Il2rg^{−/−}$ mice (Fig. 5f). Meanwhile, $Rag1^{−/−}Il7ra^{−/−}$ exhibited intermediate survival (Fig. 5f). Thus, IL-7R-independent ILC3 are functional *in vivo* to partially protect mice from *C. rodentium*.

**CCR6[+] ILC3 highly express Bcl-2 enhanced by lack of IL-7R.** Among all ILCs in the siLP of $Il7ra^{−/−}$ mice, the CCR6[+] ILC3

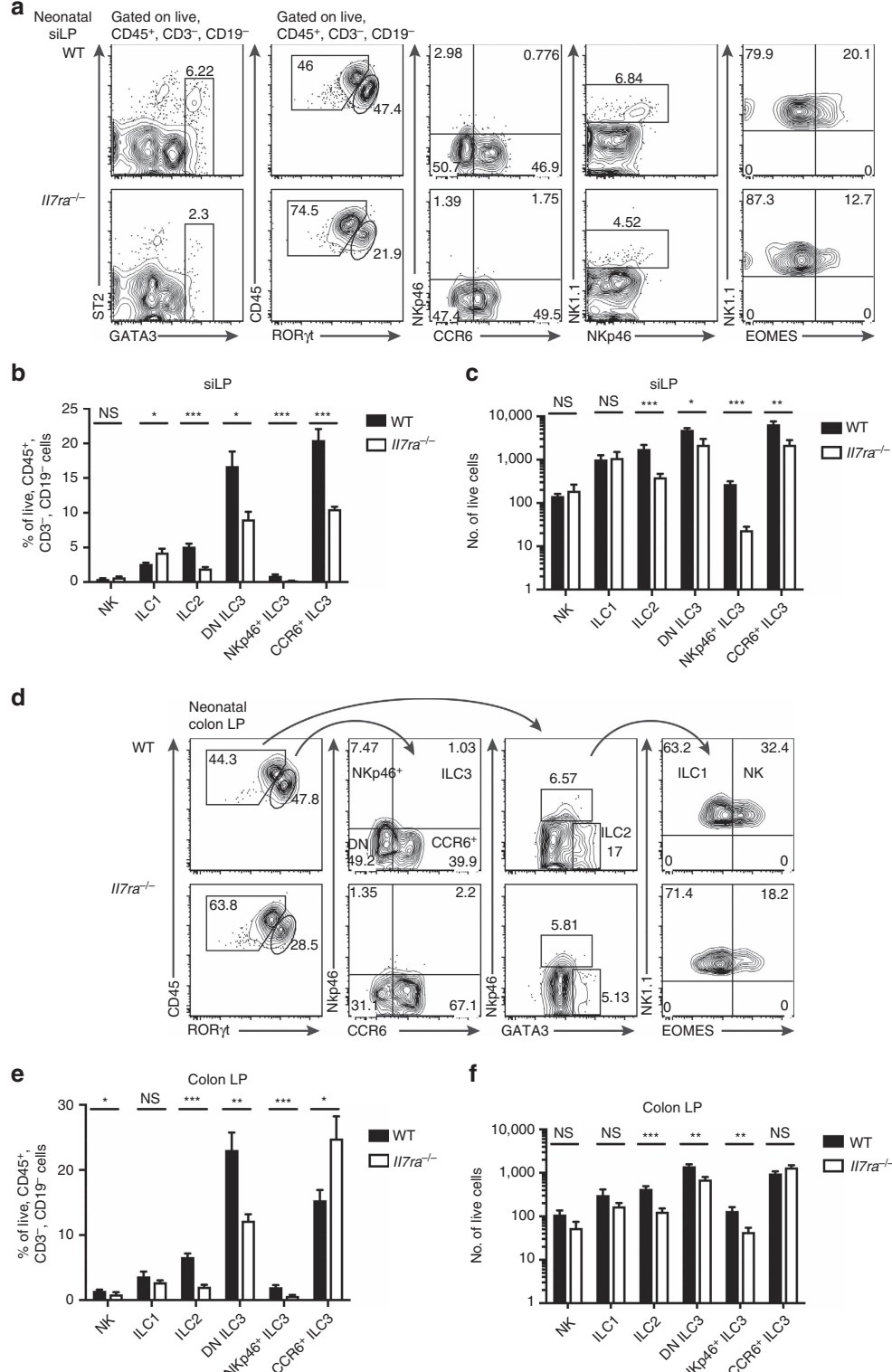

**Figure 3 | Neonatal *Il7ra*$^{-/-}$ mice have residual ILCs in siLP.** (a–c) siLP and (d–f) colon LP lymphocytes were isolated from *Il7ra*$^{-/-}$ or WT control mice on the day of birth and indicated ILC subsets assessed. (a,d) Representative flow plots of (a) siLP and (d) colon LP. (b,e) Frequency and (c,f) total number of ILC subsets from (b,c) siLP and (e,f) colon LP. Error bars mean ± s.e.m. Data represent *n* = 8 mice from 2 litters per genotype. *$P$<0.05, **$P$<0.01, ***$P$<0.001, two-tailed Mann–Whitney test.

subset is the most abundant (Fig. 1a–c). To determine whether there were any unique factors expressed by CCR6$^+$ ILC3 that could explain their relative preservation in *Il7ra*$^{-/-}$ mice, we performed microarray analysis on ILC3 isolated from WT mice. Comparisons between ILC3 subsets revealed that the CCR6$^+$

ILC3 had the greatest number of 'uniquely' expressed transcripts (*n* = 54), defined as transcripts significantly upregulated at least 2-fold compared to other ILC3 subsets (Fig. 6a and Supplementary Table 1). Because CCR6$^+$ ILC3 and NKp46$^+$ ILC3 have the greatest difference in frequency among ILC3 in

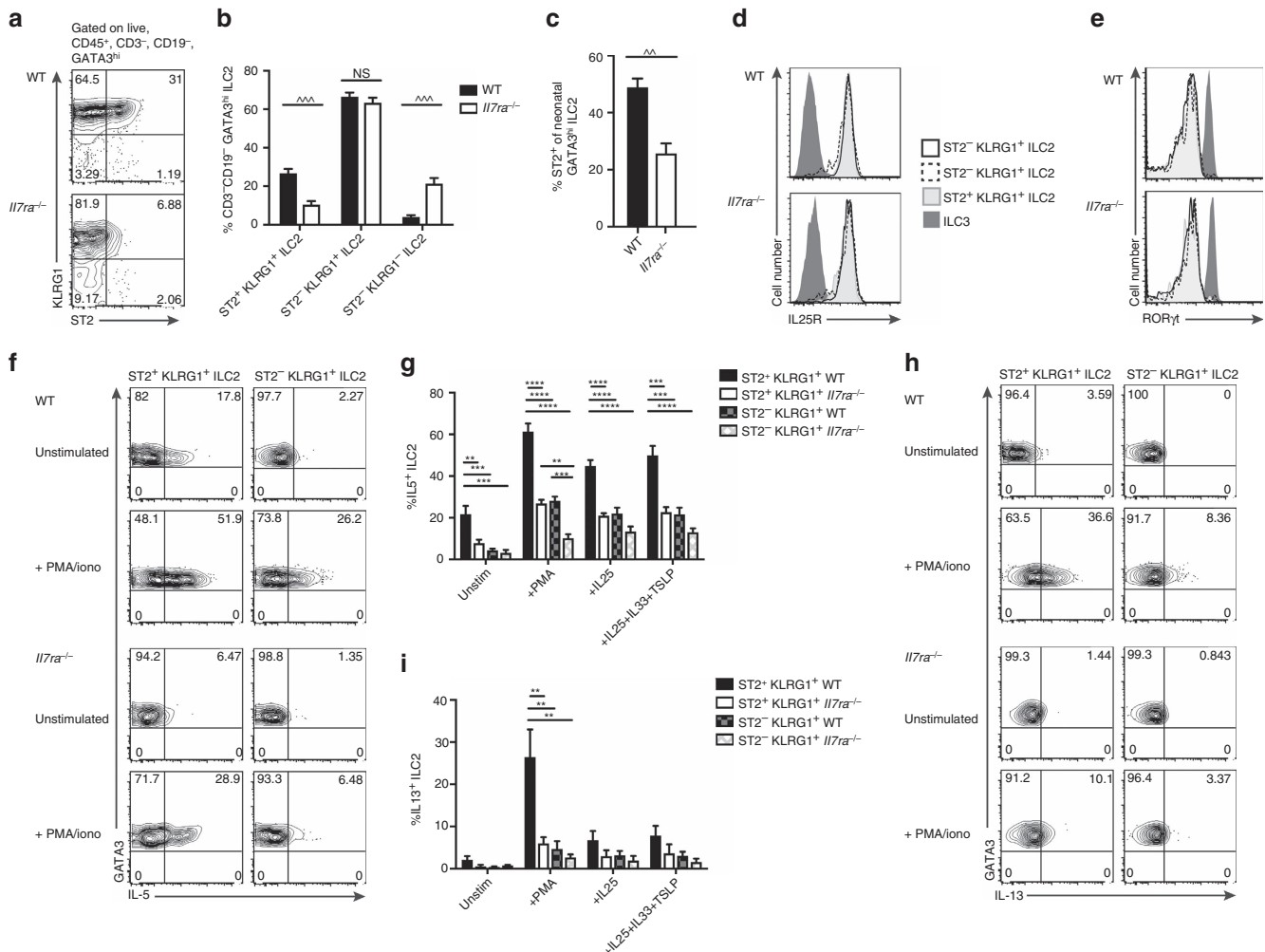

**Figure 4 | Residual *Il7ra*$^{-/-}$ ILC2 are predominantly ST2$^-$.** (**a**) Representative flow plots and (**b**) quantification of ST2$^+$KLRG1$^+$, ST2$^-$KLRG1$^+$, and ST2$^-$KLRG1$^-$ cells among GATA3$^{hi}$ ILC2 in the siLP of adult *Il7ra*$^{-/-}$ or WT control mice. (**c**) Quantification of ST2$^+$ cells among GATA3$^{hi}$ ILC2 in the siLP of neonatal *Il7ra*$^{-/-}$ or WT control mice. (**d,e**) Representative histograms of siLP ILC2 (**d**) IL-25R or (**e**) ROR$\gamma$t expression, compared to ILC3, from *Il7ra*$^{-/-}$ or WT control mice. Traces correspond to legend between the graphs. (**f**) Representative flow plots of IL-5 production by ST2$^+$ KLRG1$^+$ and ST2$^-$ KLRG1$^+$ ILC2 cells at rest or after stimulation with PMA/ionomycin, from *Il7ra*$^{-/-}$ or WT control mice. (**g**) Quantification of IL-5 production by ST2$^+$ KLRG1$^+$ and ST2$^-$ KLRG1$^+$ ILC2 cells at rest or after stimulation with PMA/ionomycin, IL-25, or IL-25 + IL-33 + TSLP, from *Il7ra*$^{-/-}$ or WT control mice. (**h**) Representative flow plots and (**i**) quantification of IL-13 production as in **f,g**. Error bars mean ± s.e.m. Data represent (**a–c**) n = 8 mice per genotype from four independent experiments or (**d–i**) n = 4 mice per genotype from two independent experiments. ^^P < 0.01, ^^^P < 0.001, two-tailed Mann–Whitney test. **P < 0.01, ***P < 0.001, ****P < 0.0001, one-way analysis of variance with Tukey's test for multiple comparisons.

*Il7ra*$^{-/-}$ mice, we next directly compared these subsets (Fig. 6b). We found that CCR6$^+$ ILC3 expressed the anti-apoptotic transcript *Bcl2* at higher levels than NKp46$^+$ ILC3 (1.965-fold, P = 0.011, Student's *t*-test; Fig. 6b).

We next tested whether Bcl-2 expression was different at the protein level in ILC3 from WT and *Il7ra*$^{-/-}$ mice (Fig. 6c). Validating our array data, we found that WT CCR6$^+$ ILC3 highly expressed Bcl-2, while NKp46$^+$ and DN ILC3 expressed lower levels (Fig. 6c). Surprisingly, in *Il7ra*$^{-/-}$ mice, Bcl-2 expression also increased in DN ILC3; NKp46$^+$ ILC3 had lowest expression, followed by intermediate expression in DN ILC3, and high expression in CCR6$^+$ ILC3 (Fig. 6c). Bcl-2 expression was significantly higher in both *Il7ra*$^{-/-}$ DN ILC3 and CCR6$^+$ ILC3 compared to WT control, but not in any other siLP ILCs (Fig. 6d). $\gamma_c$ cytokines regulate both proliferation and survival[8,39], therefore, we asked whether there were also any changes in ILC proliferation measured by Ki67 expression (Fig. 6e). Unlike Bcl-2 expression, Ki67 expression was not altered by IL-7R deficiency in CCR6$^+$ ILC3. (Fig. 6e). Instead,

more Ki67 was expressed in *Il7ra*$^{-/-}$ ILC1 and NK cells than WT cells, consistent with their relative abundance within *Il7ra*$^{-/-}$ ILC. Both CCR6$^+$ ILC3 and DN ILC3 were weakly proliferative in WT and *Il7ra*$^{-/-}$ mice, consistent with the long half-life[26] and radioresistance[40] previously reported for ILC3. Thus, IL-7R-deficiency paradoxically increases anti-apoptotic machinery in CCR6$^+$ ILC3 and, to a lesser extent, DN ILC3.

**IL-15 supports ILC survival *in vitro*.** Given the presence of all ILC subsets in the siLP of *Il7ra*$^{-/-}$ mice but their dependence on $\gamma_c$ signalling, we hypothesized that another $\gamma_c$ cytokine sustained these cells. In a screen of all $\gamma_c$ cytokines across small intestine, colon, and lung, we noted that tissue *Il15* levels were consistently high compared with other $\gamma_c$ cytokines, particularly in the GI tract (Supplementary Fig. 1A–C). As IL-15 supports the development of NK cells[9,10] and ILC1 (ref. 13), we hypothesized that this cytokine may partially compensate for the loss of IL-7. To test this, we first asked if ILC possessed the machinery to

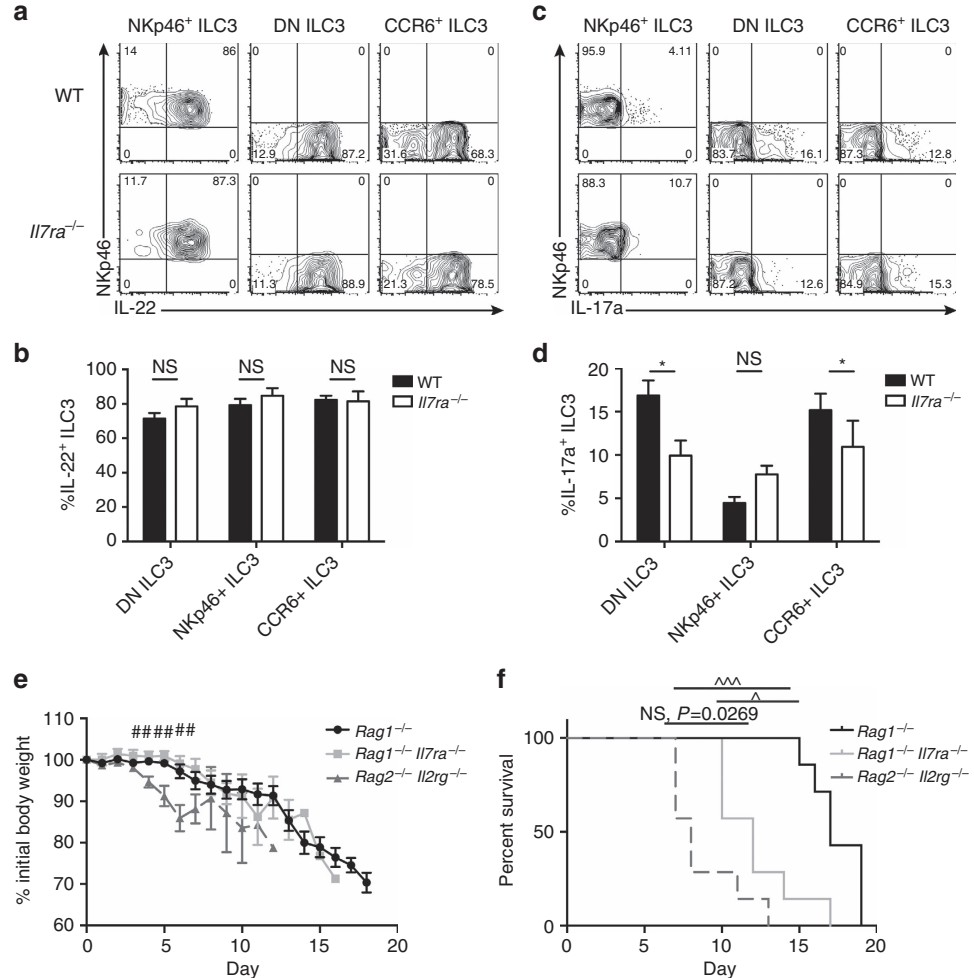

**Figure 5 | Functional *Il7ra*$^{-/-}$ ILC3 partially protect from *C. rodentium* infection.** (**a**) Representative flow plots and (**b**) quantification of IL-22 production from NKp46$^+$, DN, and CCR6$^+$ ILC3 subsets after stimulation with IL1β and IL-23. ILC3 were electronically gated on CD3$^-$ CD19$^-$ CD45$^{lo}$ Thy1$^+$ and subdivided by NKp46 and CCR6. (**c**) Representative flow plots and (**d**) quantification of IL-17a production from ILC3 subsets, as in **a**. (**e**) Weight loss and (**f**) survival of *Rag1*$^{-/-}$, *Rag1*$^{-/-}$ *Il7ra*$^{-/-}$ and *Rag2*$^{-/-}$ *Il2rg*$^{-/-}$ mice after infection with $2 \times 10^9$ c.f.u. *C. rodentium*. Error bars mean ± s.e.m. Data represent (**a–d**) $n = 6$ mice per genotype from 3 independent experiments or (**e,f**) $n = 7$ mice per genotype from 2 independent experiments. *$P < 0.05$, two-tailed Mann–Whitney test. $^{\#\#}P < 0.01$, one-way analysis of variance. $^{\wedge}P < 0.0167$, $^{\wedge\wedge\wedge}P < 0.0003$, Gehan-Breslow-Wilcoxon test with Bonferroni correction for multiple comparisons.

recognize IL-15 by measuring expression of the IL-2/IL-15 receptor CD122 in the siLP (Fig. 7a) and colon LP (Fig. 7b) of WT and *Il7ra*$^{-/-}$ mice. All ILCs expressed CD122, with highest expression in NK cells, followed by ILC1, ILC3, and lowest expression in ILC2 (Fig. 7a,b). siLP *Il7ra*$^{-/-}$ ILC2 and ILC3 expressed CD122 at higher levels than WT; there was no difference between genotypes in the colon (Fig. 7a,b). We conclude that ILC have the capacity to respond to IL-15, which may be higher in the siLP of *Il7ra*$^{-/-}$ mice.

There is evidence in the literature that IL-15 can sustain ILC, as mouse ILC3 have previously been cultured in IL-7 (20 ng ml$^{-1}$) and IL-15 (50 ng ml$^{-1}$)[18]. To expand these prior results and to comprehensively test if IL-7 and IL-15 have different capacities to sustain ILCs, we sorted ILC1/NK cells, CCR6$^+$ ILC3, CCR6$^-$ ILC3 (containing the T-bet$^{lo/+}$ lineage of DN and NKp46$^+$ ILC3), and ILC2 from WT and *Il7ra*$^{-/-}$ mice, and cultured equal numbers of cells from each genotype for 2 days in IL-7 (20 ng ml$^{-1}$), IL-15 (20 and 50 ng ml$^{-1}$), or medium alone. We then assessed survival with 7aad (Fig. 7c–f). We found that IL-15 significantly increased the percentage of surviving NK/ILC1 (Fig. 7c), ILC2 (Fig. 7d), and CCR6$^-$ ILC3 (Fig. 7e) from both

WT and *Il7ra*$^{-/-}$ mice, while IL-7 only significantly increased the survival of WT ILCs (Fig. 7c–e). High dose IL-15 better supported ILC survival than low dose except for WT ILC2, which was not statistically significant (Fig. 7c–e). We detected no difference in the ability of IL-15 to sustain ILC survival between genotypes (Fig. 7c–e). The exception was CCR6$^+$ ILC3, from which *Il7ra*$^{-/-}$ cells survived significantly better in low dose IL-15 than WT cells (Fig. 7f). In general, CCR6$^+$ ILC3 were resistant to γ$_c$ cytokine depletion, and exhibited high survival in medium, with no significant differences in either genotype between culture conditions (Fig 7f).

Although IL-15 sustained ILC *in vitro*, it was not equivalent to IL-7. WT NK/ILC1 survived significantly better in IL-15 than IL-7 (Fig. 7c), and grew substantially larger than cells cultured in IL-7 (Fig. 7g). In contrast, WT ILC2, CCR6$^-$ ILC3 and CCR6$^+$ ILC3 grew larger in size in IL-7 than in IL-15 (Fig. 7h–j). For WT and *Il7ra*$^{-/-}$ CCR6$^+$ ILC3, which showed no significant difference in survival between any culture conditions, IL-15 also supported larger cells than medium alone (Fig. 7j). We conclude that IL-15 sustains ILC survival but signals are qualitatively different from IL-7.

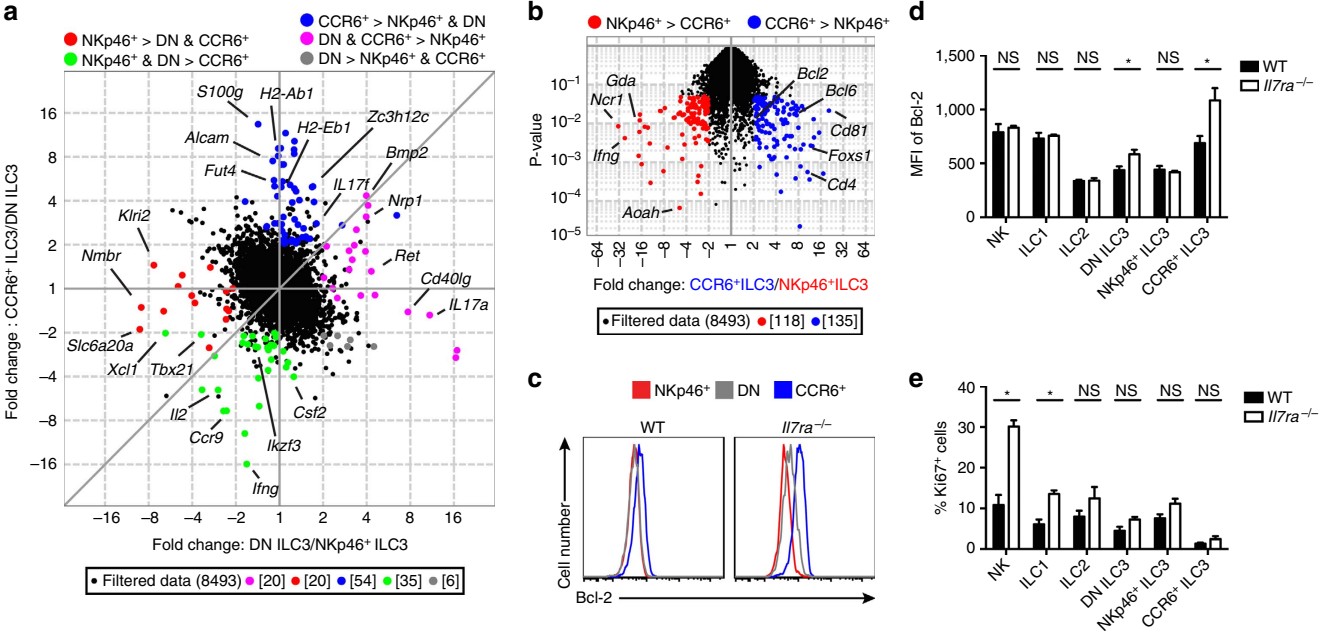

**Figure 6 | CCR6+ ILC3 highly express Bcl-2 enhanced by lack of IL-7R.** (**a**) Plot comparing the fold change of CCR6+ ILC3 and DN ILC3 to the fold-change of DN ILC3 and NKp46+ ILC3 depicts transcripts differentially expressed among ILC3 subsets. (**b**) Volcano plot comparing CCR6+ ILC3 to NKp46+ ILC3. (**a,b**) Coloured circles represent transcripts expressed ≥2-fold with *P*-value ≤0.05 in comparisons matching the colour code above the plot. (**c**) Representative histograms of Bcl-2 protein expression measured by flow cytometry among ILC3 subsets in WT and *Il7ra*−/− mice. (**d**) Mean fluorescent intensity (MFI) of Bcl-2 and (**e**) percentage of Ki67+ cells among indicated siLP ILC subsets in WT and *Il7ra*−/− mice, measured by flow cytometry. Error bars mean ± s.e.m. Data represent (**a,b**) 3 pooled mice each or (**c–e**) *n* = 5 mice per genotype from (**a–e**) 2 independent experiments. *P<0.05, two-tailed Mann–Whitney test.

**IL-15 partially compensates for IL-7 in siLP ILC development.** To directly test the hypothesis that IL-15 sustained ILCs in the absence of IL-7, we generated *Il7ra*−/−*Il15*−/− mice. We then assessed the frequency and total number of ILCs in the siLP (Fig. 8a,b; Supplementary Fig. 2A), colon LP (Fig. 8c,d), lung (Fig. 8e,f), and adipose tissue (Fig. 8g,h) of these mice compared to WT, *Il15*−/−, *Il7ra*−/−, and *Rag2*−/−*Il2rg*−/− mice using our intracellular TF strategy (Fig. 1a,g,j). We also assessed restricted ILC progenitors from the bone marrow that could be detected in the absence of CD127 expression, rNKP (Fig. 8i,j) and ILC2P (Fig. 8k,l).

We first turned our attention to ILC1 and NK cells. Between WT and *Il15*−/− mice, colon ILC1 were significantly reduced in frequency and siLP NK cells and ILC1 had reduced total numbers (Fig. 8a–d). All other populations trended towards decreased frequencies and total numbers. Between WT and *Il7ra*−/−*Il15*−/− mice, colon ILC1 were also significantly reduced in frequency, and all ILCs were significantly reduced in total numbers (Fig. 8a–d; Supplementary Fig. 2A). There were no differences between *Il7ra*−/−*Il15*−/− and *Rag2*−/−*Il2rg*−/− mice (Fig. 8a–d). Direct comparisons of NK cells and ILC1 between *Il15*−/− and *Il7ra*−/−*Il15*−/− were not statistically significant; we did note that the magnitude of depletion appeared greater in *Il7ra*−/−*Il15*−/− siLP ILC1, colon ILC1, and colon NK cells, but not for siLP NK cells compared to *Il15*−/− mice (Fig. 8b,d). To determine whether differences in ILC1 and NK cell generation occurred at a progenitor stage or in the periphery, we next assessed rNKPs, which PLZF fate-mapping studies have shown include both ILC1 and NK cell progenitors[41]. These cells were equally reduced in frequency (Fig. 8i) and number (Fig. 8j) in *Il15*−/−, *Il7ra*−/−*Il15*−/−, and *Rag2*−/−*Il2rg*−/− mice, but not *Il7ra*−/− mice. We conclude that IL-15 primarily supports ILC1 and NK cells, but that IL-7 may have a somewhat greater, though variable, role in ILC1 and colon NK cell in the periphery,

consistent with the higher expression of IL-7R by these cells (Fig. 2a,b).

For ILC2, we again found that these cells were selectively preserved in the siLP between WT and *Il7ra*−/− mice (Fig. 8a–f). In *Il7ra*−/−*Il15*−/− and *Rag2*−/−*Il2rg*−/− siLP, ILC2 frequencies were significantly reduced when compared with *Il7ra*−/− mice (Fig. 8a). This resulted in a reproducible trend towards reduced total numbers of ILC2, that was significant between *Il7ra*−/− and *Il7ra*−/−*Il15*−/− mice in a separate set of experiments comparing only WT, *Il7ra*−/−, and *Il7ra*−/−*Il15*−/− genotypes (Supplementary Fig. 2B). In colon, lung, and adipose tissue where ILC2 were already minor populations in *Il7ra*−/− mice, we noted no further reductions in frequency or total number in *Il7ra*−/−*Il15*−/− mice (Fig. 8a–g). There were no differences between WT and *Il15*−/− generation of ILC2 in any organ.

To determine whether IL-15 supported ILC2 in the bone marrow or the periphery, we next assessed ILC2P across all genotypes. We found that ILC2P frequency and total number were significantly reduced in *Il7ra*−/− mice compared to WT and *Il15*−/−, but there was no further reduction in *Il7ra*−/−, *Il7ra*−/−*Il15*−/− or *Rag2*−/−*Il2rg*−/− genotypes (Fig. 8k,l). We conclude that IL-15 most likely supports ILC2 survival in the periphery.

Last, we assessed ILC3. CCR6+ ILC3 were the most abundant subset in *Il7ra*−/− siLP compared to WT, while other ILC3 were significantly reduced in frequency and number but still present in the siLP (Fig. 8a,b). Interestingly, we noted that CCR6+ ILC3 were also the most abundant population in *Il7ra*−/−*Il15*−/− and *Rag2*−/−*Il2rg*−/− mice, though significantly reduced in frequency and number compared with WT and *Il7ra*−/− mice (Fig. 8a,b; Supplementary Fig. 2A,B). Therefore, we tested Bcl-2 in ILC3 across the five genotypes. There were no differences in Bcl-2 expression between *Il15*−/− and WT ILC3.

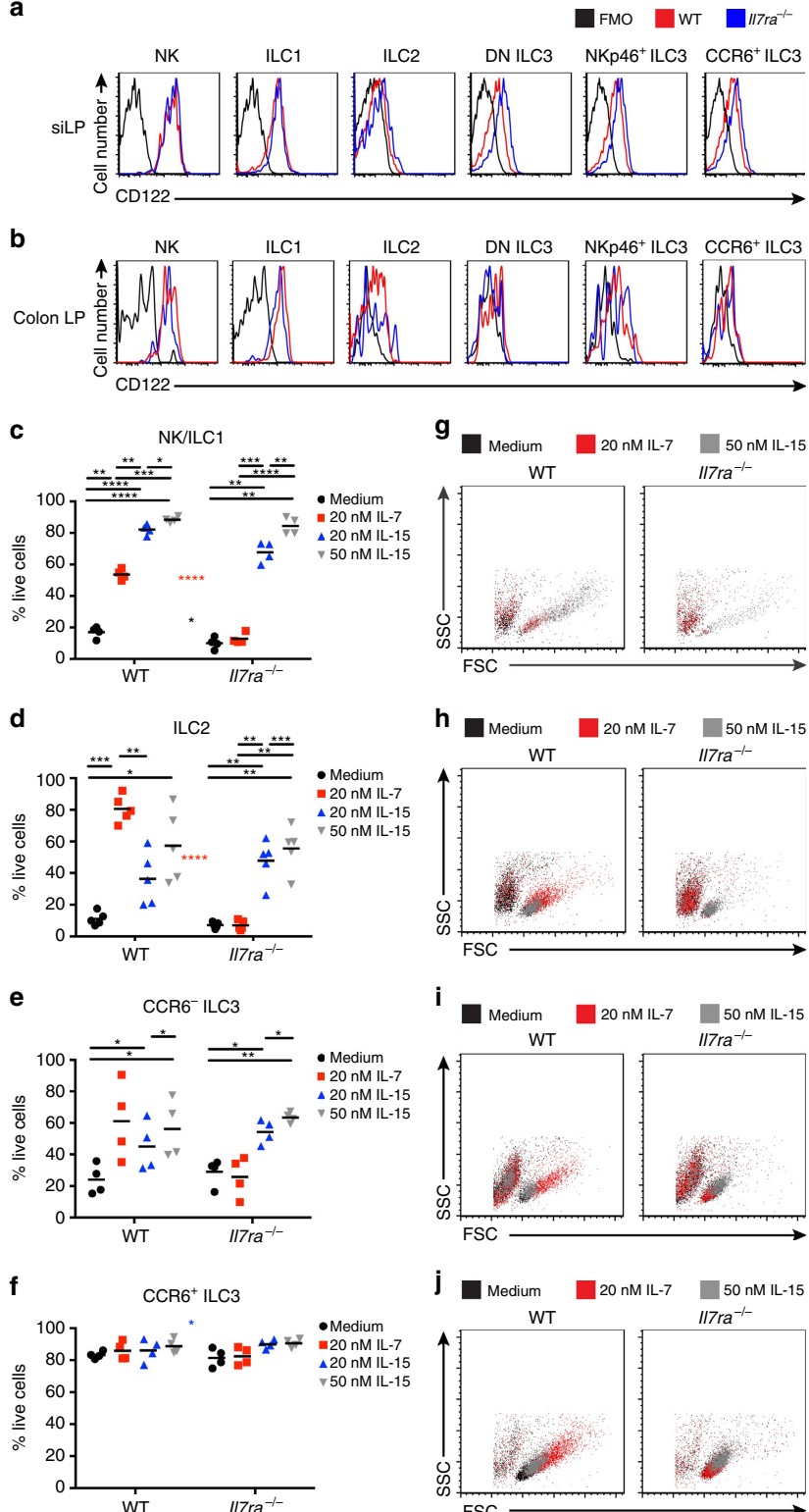

**Figure 7 | All ILCs express CD122 and are sustained by IL-15 *in vitro*.** (**a,b**) Representative histograms of CD122 expression by ILCs from (**a**) siLP and (**b**) colon from WT and *Il7ra*$^{-/-}$ mice using biotinylated anti-CD122 with SAv-PeCy7 compared to FMO controls. Traces corresponding to the legend above A. (**c–j**) ILCs were concurrently sorted from the siLP of WT and *Il7ra*$^{-/-}$ mice and were cultured in the indicated culture conditions for two days before assessing survival with 7aad. Data are shown for (**c,g**) NK/ILC1 (**d,h**) ILC2 (**e,i**) CCR6$^-$ ILC3 and (**f,j**) CCR6$^+$ ILC3. (**c–f**) Quantification of the percentage of 7aad$^-$ cells after culture. Coloured asterisks between graphs correspond to significant differences between genotypes, colour-coded to the key to the right of each graph. (**g–j**) Representative, stacked flow scatter plots from culture conditions, colour-coded to the legend above each graph. Data represent (**a,b**) $n = 6$ mice, (**c–e–g,i,j**) $n = 4$ mice or (**d,h**) $n = 5$ mice per genotype from (**a–c,e–g,i,j**) 3 or (**d,h**) 4 independent experiments. Error bars mean ± s.e.m. *$P < 0.05$, **$P < 0.01$, ***$P < 0.001$, ****$P < 0.0001$. RM one-way analysis of variance with Geisser-Greenhouse correction and Tukey's test for multiple comparisons.

As in $Il7ra^{-/-}$ mice, we found that Bcl-2 expression was also higher than WT in $Il7ra^{-/-} Il15^{-/-}$ DN and CCR6$^+$ ILC3, but not in NKp46$^+$ ILC3 (Supplementary Fig. 2C). DN Bcl-2

expression was comparable between $Il7ra^{-/-}$ and $Il7ra^{-/-} Il15^{-/-}$ mice, but was more highly expressed by $Il7ra^{-/-} Il15^{-/-}$ CCR6$^+$ ILC3 than $Il7ra^{-/-}$ CCR6$^+$ ILC3. We further

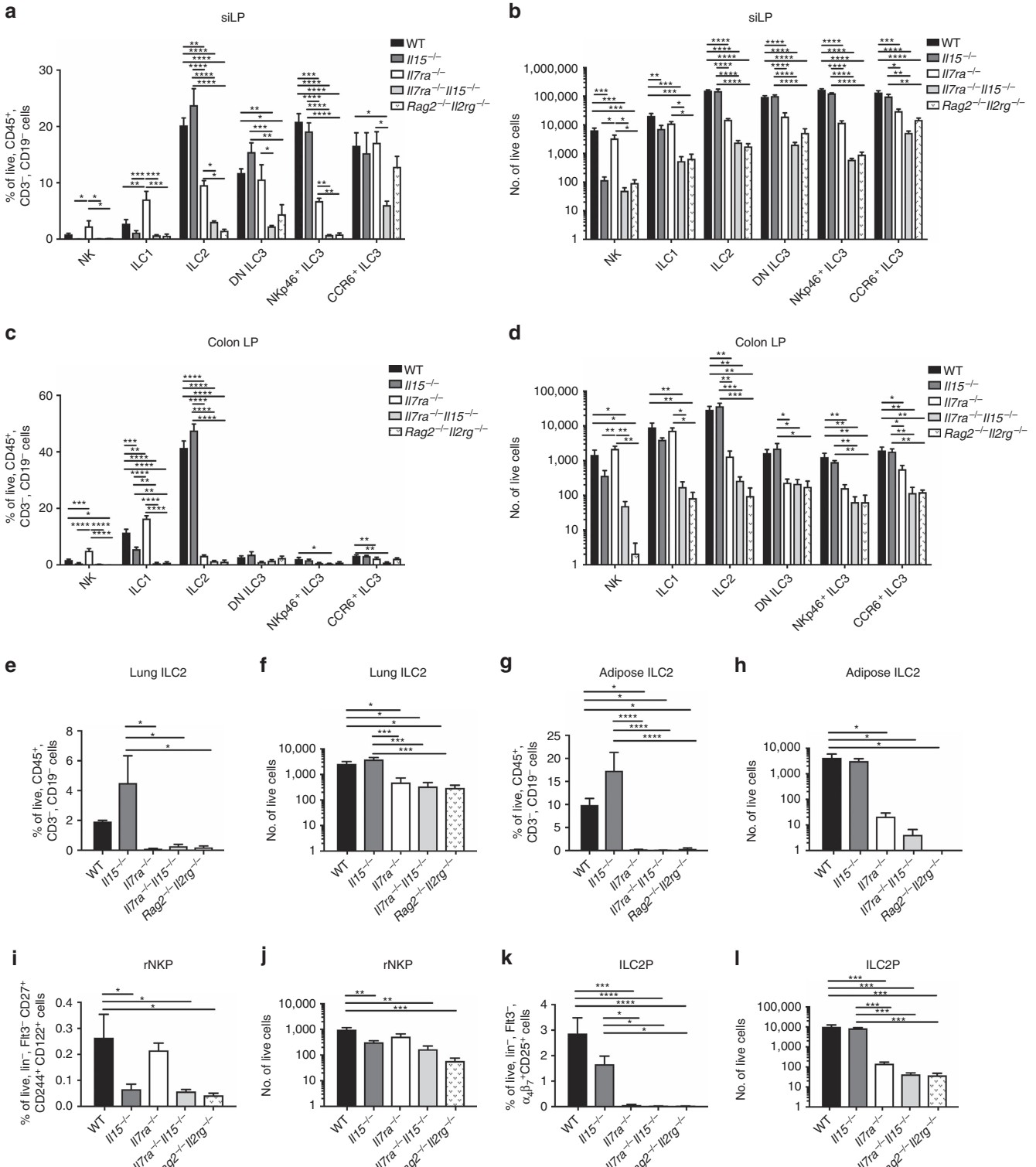

**Figure 8 | $Il7ra^{-/-} Il15^{-/-}$ mice generate fewer ILCs than $Il7ra^{-/-}$ mice.** (**a,b**) siLP, (**c,d**) colon LP, (**e,f**) lung and (**g,h**) gonadal adipose tissue ILCs or (**i–l**) the bone marrow restricted ILC progenitors from two tibias (**i,j**) rNKPs (Lin$^-$ Flt3$^-$ CD27$^+$ CD244$^+$ CD122$^+$) and (**k,l**) ILC2P (Lin$^-$ Flt3$^-$ $\alpha_4\beta_7^+$ CD25$^+$) were directly compared between WT, $Il15^{-/-}$, $Il7ra^{-/-}$, $Il7ra^{-/-} Il15^{-/-}$, and $Rag2^{-/-} Il2rg^{-/-}$ mice. (**a,c,e,g,i,k**) Frequency and (**b,d,f,h,j,l**) total number of indicated ILC or ILC progenitor subsets. (**i–l**) Lineage included anti- B220, CD3, CD4, CD8, CD11b, CD11c, CD19, GR1, NK1.1 and Ter119. Data represent (**a–l**) $n = 4$ mice per genotype from two independent experiments. Error bars mean ± s.e.m. *$P < 0.05$, **$P < 0.01$, ***$P < 0.001$, ****$P < 0.0001$, one-way analysis of variance with Tukey's test for multiple comparisons.

noted that Bcl-2 expression was different between $Il7ra^{-/-}$ $Il15^{-/-}$ and $Rag2^{-/-} Il2rg^{-/-}$ ILC3, with lower Bcl-2 expression in both subsets by $Rag2^{-/-} Il2rg^{-/-}$ ILC3 reaching back to WT baseline in DN ILC3 and to $Il7ra^{-/-}$ levels in CCR6$^+$ ILC3 (Supplementary Fig. 2C). Collectively, these data suggest that Bcl-2 expression in DN ILC3 and CCR6$^+$ ILC3 is supported by a $\gamma_c$ cytokine other than IL-15 or IL-7, while expression in CCR6$^+$ ILC3 is also somewhat $\gamma_c$ independent. Yet, higher Bcl-2 levels did not rescue $Il7ra^{-/-} Il15^{-/-}$ or $Rag2^{-/-} Il2rg^{-/-}$ ILC3 numbers. We conclude that IL-15 partially compensates for IL-7 in the development and/or maintenance of siLP ILC3 in the absence of IL-7R, independent of direct effects on Bcl-2.

## Discussion

The signals that phenotypically condition and maintain tissue-resident ILCs are emerging. Here we systematically assessed the role of IL-7R in ILCs, and demonstrated that substantial numbers of functionally capable ILC2 and ILC3 persist in the siLP of adult $Il7ra^{-/-}$ mice, but not in other organs. We also show for the first time that ILC2 and ILC3 persist in neonatal $Il7ra^{-/-}$ mice. Though these data do not rule out a role for changes to the microbiota or tissue microenvironment in the phenotype of adult $Il7ra^{-/-}$ mice, they suggest that the generation of IL-7R-independent cells is unlikely to be a purely reactive process but rather that the requirement for IL-7R is incomplete. However, environmental changes may have some effect. For example, we find that the dependency of IL-7 in colon NK cells was more marked in some experiments than others. Thus, the extent to which ILCs depend or do not depend on IL-7 in the periphery is context-dependent.

ILC2 persisted in the greatest frequency and number in the siLP of $Il7ra^{-/-}$ mice compared to other commonly studied ILC2-harbouring organs, but their phenotype and functions were changed. We found that WT siLP ILC2 include ST2$^+$KLRG1$^+$, ST2$^-$KLRG1$^+$, and immature ST2$^-$KLRG1$^-$ subsets[20]. $Il7ra^{-/-}$ mice had altered ratios of ILC2 subsets, with fewer expressing the ST2$^+$KLRG1$^+$ phenotype and more expressing the immature ST2$^-$KLRG1$^-$ phenotype. Because ILC2 are very rare in non-siLP $Il7ra^{-/-}$ organs, the persistence of IL-25 induced ST2$^-$KLRG1$^+$ iILC2 in $Il7ra^{-/-}$ lung has been used to argue that these cells have different developmental origins from other ILC2 (ref. 35). Our data suggest that $Il7ra^{-/-}$ mice can support ST2$^-$ ILC2 regardless of inflammation, but perhaps that peripheral ILC2 maturation and/or survival is impaired in the absence of IL-7.

ST2$^-$ cells first began to preferentially accrue in $Il7ra^{-/-}$ neonates, suggesting a cell-intrinsic role for IL-7R in the development ST2$^+$ ILC2. Yet, functional distinctions between WT ST2$^+$ and ST2$^-$ ILC2, as well as the functional capacity of $Il7ra^{-/-}$ ILC2, remained unclear. In analyses of IL-5 and IL-13 expression, we found that only WT ST2$^+$KLRG1$^+$ substantially produced IL-13, and that WT ST2$^-$KLRG1$^+$ ILC2 as well as all IL-7R deficient ILC2 were functionally impaired compared to these cells. Interestingly, investigators have recently shown that epithelial tuft cells are markedly reduced in $Il7ra^{-/-}$ mice[32]. These cells produce IL-25, inducing ILC2 to produce IL-13, which then feeds back on epithelial stem cells to increase tuft cell generation[32–34]. Therefore, our data suggests that the ST2$^+$ subset of ILC2 may uniquely regulate tuft cell generation and that there is a break in the tuft cell cycle in IL-7R deficient mice.

The roles of IL-7R in ILC3 function has been somewhat controversial. A previous report demonstrated that $Il7^{-/-}$ NKp46$^+$ ILC3 were unable to generate IL-22 (ref. 19), while a recent report showed that bulk $Il7^{-/-}$ ILC3 were able to produce

significantly more IL-22 after stimulation with IL-23 in vitro[21]. To assess both IL-17 and IL-22 production, we stimulated cells with IL-1β and IL-23, and found that all subsets of ILC3 were equally capable to produce IL-22, but had moderately impaired ability to produce IL-17a. As our results for IL-22 were intermediate between earlier reports, perhaps reflecting differences in stimulation conditions, we also tested the function of these cells in vivo for the first time by generating $Rag1^{-/-}$ $Il7ra^{-/-}$ mice and challenging them with C. rodentium. $Il7ra^{-/-} Rag1^{-/-}$ had an intermediate phenotype between $Rag1^{-/-}$ and $Rag2^{-/-} Il2rg^{-/-}$, most likely reflecting intact ILC3 production of IL-22 from reduced numbers of cells. Collectively, we conclude that IL-7R deficient ILC3 are at least partially functional, both in vitro and in vivo.

What signals compensate for IL-7 and facilitate the relative preservation of ILC2 and ILC3 in the siLP of $Il7ra^{-/-}$ mice? We hypothesized that the $\gamma_c$ cytokine IL-15 may be the compensatory signal, as this cytokine: maintains NK cells, ILC1, and other intestinal lymphocyte populations[42]; provides compensatory signals to other innate lymphocyte subsets in the absence of IL-7 (ref. 43); and, like IL-7, acts through STAT5 to induce pro-survival Bcl-2, which was upregulated in $Il7ra^{-/-}$ CCR6$^+$ and DN ILC3[8,39]. To test this hypothesis, we first established that all ILCs expressed CD122, allowing them to signal through IL-15. Indeed, siLP $Il7ra^{-/-}$ ILC2 and ILC3 expressed greater amounts of CD122 than their WT counterparts, a change we did not detect between genotypes in colon ILC2 and ILC3. In culture, IL-15 could equally increase the survival of both siLP WT and $Il7ra^{-/-}$ ILCP-derived ILC in spite of changes in CD122 expression between genotypes. In vivo, organ-specific upregulation of CD122 in ILC2 and ILC3 could further support the enhanced maintenance of ILCs. Thus, IL-7 and IL-15 could both support WT ILCP-derived ILC survival. In contrast to other ILCs, CCR6$^+$ ILC3 were long-lived in culture and survived independent of $\gamma_c$ signalling for two days, consistent with a prior report[44].

Residual IL-7R deficient ILCs were sustained by IL-15; when IL-15 was also deleted, residual ILC were depleted to levels found in $Rag2^{-/-} Il2rg^{-/-}$ mice. Given that IL-7R is a critical cell-surface marker for ILC progenitors that is essentially missing in our experiments, we were unable to determine whether IL-15 sustained proximal restricted progenitors including CHILP and ILCP. However, we were able to evaluate ILC2P and rNKPs, and found no additional depletion of these progenitors between $Il7ra^{-/-}$ and $Il7ra^{-/-} Il15^{-/-}$ mice. While these data do not rule out reduced fitness of residual restricted progenitors in the absence of both IL-7R and IL-15, considering our CD122 expression and in vitro survival data, we conclude that IL-7 and IL-15 mutual compensation for ILC1, NK cells and ILC2 most likely acts in the periphery rather than at a progenitor stage.

Unlike other ILCs, selective restricted progenitors for ILC3 have not been identified in the adult bone marrow. In the periphery, we found that DN and CCR6$^+$ ILC3 frequencies correlated with Bcl-2 expression in $Il7ra^{-/-}$ mice. Yet, this mechanism did not explain the role of IL-15 in ILC3 maintenance because DN ILC3 Bcl-2 upregulation was completely $\gamma_c$-dependent, but IL-15-independent; CCR6$^+$ ILC3 Bcl-2 upregulation was partially $\gamma_c$-independent but was also fully IL-15-independent. As DN and CCR6$^+$ ILC3 also express the high affinity IL-2 receptor CD25 (ref. 17), which similarly activates STAT5 (ref. 8), IL-2 may contribute to the Bcl-2 upregulation in these cells.

Does IL-15 physiologically impact ILC2 and ILC3? Besides the unchanged ILC2 and ILC3 in IL-15 deficient mice we report here, IL-15 overexpressing transgenic mice or $Cish^{-/-}$ mice with increased IL-15 signalling do not have changed numbers of siLP

NKp46[+] ILC3 (ref. 19) or lung ILC2 (ref. 45). This suggests that at least in steady state, IL-15 has limited trophic roles for these populations. For ILC3, IL-15 can sustain both human[46] and mouse[18] cells *in vitro*, though it increases their plasticity to IFN-γ producing ILC1-like cells. Furthermore, IL-15 is substantially produced in the siLP, and unlike IL-7, is increased during epithelial damage[42,47]. In an inflammatory context when IL-7 trophic signals are limiting, IL-15 may provide signals to ILCs.

In the modern paradigm that NK cells mirror CD8[+] T cells, while ILCs parallel CD4[+] T cells[3], it is worthwhile to consider the $\gamma_c$ requirements of T cells. All naïve T cells predominantly utilize T cell receptor (TCR) signalling and IL-7 for their survival[8,39,48]. Meanwhile, similar to TCR-lacking ILCs, memory T cells are TCR independent and instead use $\gamma_c$ cytokines for survival[39,49]. Like NK cells, memory CD8[+] T cells use more IL-15, while like ILCs, memory CD4[+] T cells use more IL-7 (ref. 39). However, both CD8[+] and CD4[+] memory T cells use a combination of IL-7 and IL-15 for long term survival[8,49]. Like ILC2 and ILC3, memory CD4[+] T cells were originally considered to be IL-15 independent, because memory CD4[+] T cells are normal in IL-15-deficient mice. Now, it is recognized that memory CD4[+] T cells require both IL-15 and IL-7 for optimal fitness and long term survival in WT, but not lymphopenic, mice[39,50,51]. Collectively, these data combined with our data from *Il7ra*[−/−] and *Il7ra*[−/−]*Il15*[−/−] mice hint at a similar physiologic function of IL-15 in ILCs, which remains to be more comprehensively interrogated.

## Methods

**Mice.** *Il15*[−/−], *Rag1*[−/−] and C57BL6/J mice were purchased from the Jackson Laboratory. *Rag2*[−/−]*Il2rg*[−/−] mice were purchased from Taconic. *Il7ra*[−/−] mice were a gift from Dr. Ken Murphy. All mice were bred and maintained in a pathogen-free facility at Washington University. Age- and sex-matched animals were analysed on the day of birth or between 6 and 16 weeks of age. The WUSM Animal Studies Committee approved all experiments.

**Antibodies and flow cytometry.** Antibody information can be found in Supplementary Table 2. LIVE/DEAD Fixable Aqua was from Life Technologies. 123count eBeads were obtained from eBioscience. Fc receptors were blocked before staining with supernatant from hybridoma cells producing monoclonal antibody to CD32 (HB-197; ATCC). For intracellular transcription factor staining, the FOXP3 staining kit (eBioscience) was used. For intracellular cytokine staining, the BD Cytofix/Cytoperm kit was used. Data were acquired on a BD FACSCanto II or BD LSRFortessa and analysed with FlowJo software (Treestar).

**Cell isolation and culture.** Adult siLP, colon LP and lung immune cells were isolated using Collagenase 4 (Sigma) digestion and Percoll (GE Healthcare) gradient enrichment. Adipose tissue cells were isolated using Collagenase D (Roche). For intestinal samples, mesenteric adipose tissue, Peyer's patches, and IELs were first removed by dissection and two EDTA extraction washes. Neonatal siLP and colon LPs were processed similarly, but Peyer's patches were not removed and samples were not Percoll gradient enriched. For functional experiments, siLP cells were stimulated using recombinant cytokines or PMA/ionomycin for 3 h, followed by three additional hours in the presence of Golgi Plug (BD). For survival experiments, 4 populations of cells were sorted using a BD FACSAria from *Il7ra*[−/−] and WT mice concurrently among live, CD45[+] CD3[−] CD19[−] CD5[−] lymphocytes: NKp46[+] NK1.1[+] 'ILC1/NK', NK1.1[−] KLRG1[+] 'ILC2', NK1.1[−] KLRG1[−] CCR6[+] 'CCR6[+] ILC3' and NK1.1[−] KLRG1[−] CCR6[−] CD45[lo] Thy1[+] 'CCR6[−] ILC3s'. Four wells containing equal numbers of ILCs (1,000–5,000 cells per well) from each genotype were then plated in 96-well round bottom plates and cultured in the indicated cytokine or medium alone for 2 days. 7aad staining was performed in the Annexin 5 staining buffer for 15 min at room temperature (BD). All cytokines (Peprotech, R&D) were used at 10 ng ml[−1], unless otherwise noted.

**Citrobacter rodentium infection.** Cohoused experimental mice were fasted for 4 h, then gavaged with $2 \times 10^9$ c.f.u. *C. rodentium* in 100 µl PBS. Weight loss was measured at the same time daily and normalized to the mass recorded on the day of infection.

**Microarray analysis.** ILC3 subsets were sorted from the siLP of three pooled WT donors. Cells were gated on CD3[−] CD19[−] CD45[lo] Thy1[+] ILC3s, then subdivided into three subsets using CCR6 and NKp46 (resulting in CCR6[+], NKp46[+] and DN

ILC3). RNA was isolated using the RNeasy Micro Kit (Qiagen). The Genome Technology Access Center (GTAC) at Washington University amplified with Ovation PicoSL (NuGEN) and hybridized the product to Affymetrix Mouse Gene 1.0 ST arrays. Analysis was carried out as previously described[17]. Briefly, CEL files were normalized with AffySTExpressionFileCreator module of GenePattern using RMA. Differences in gene expression were identified using the Multiplot Studio function of GenePattern[17] (Broad Institute), from a filtered subset of genes with coefficients of variation less than 0.1 in all samples and expression of at least 120 relative units in one subset by the class mean function. Gene expression was considered 'unique' or 'shared' if expression was greater than twofold and *P* values < 0.05 (Student's *t*-test) in the indicated subset. Graphs were produced in Multiplot Studio.

**Quantitative reverse transcriptase–PCR.** Lower lobe of left lung, terminal ileum, and proximal colon immediately distal to the cecum were dissected and immediately preserved in RNA*later* (Ambion). To extract RNA, samples were placed into RNeasy Mini RLT buffer (Qiagen) containing 1% β-mercaptoethanol (Fisher) and homogenized using a MagNA Lyser (Roche). RNA was extracted using the RNeasy Mini kit (Qiagen) and on-column digestion with RNase-free DNase (Qiagen) per manufacturer instructions. Purified RNA was reverse transcribed with oligo(dT)$_{20}$ primer using the SuperScript III first strand synthesis system (Invitrogen). Resultant cDNA was diluted in nuclease-free water and used for real-time PCR using iTaq Universal SYBR Green Supermix (Bio-Rad) on a LightCycler 96 thermocycler (Roche). Primers are found in Supplementary Table 3.

**Statistics.** Prism 7 (GraphPad Software) was used for all statistical analyses except for microarray analysis, which was performed in the Multiplot Studio function of GenePattern. All graphical data show mean ± s.e.m.

**Data availability.** Microarray data that support the findings of this study have been deposited in GEO with the primary accession code GSE92693. All other data that support the findings of this study are available from the corresponding author on reasonable request.

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

## Acknowledgements

Supported by the US National Institutes of Health (U01AI095542 and R21CA16719 to M.C. and 1F30DK107053-01 to M.L.R.).

## Author contributions

M.L.R., J.K.B., W.S. and T.K.U. performed experiments. M.L.R. analysed data. M.L.R. and S.G. generated and maintained mice. M.L.R. and M.C. designed studies. M.L.R. and M.C. wrote the paper.

## Additional information

**Competing financial interests:** The authors declare no competing financial interests.

