## [Peer Review File · Nature Communications]

Reviewers' comments:

Reviewer #1 (Remarks to the Author):

Robinette et al. examined the role of IL-7R signaling in ILC differentiation and reported in this paper that unlike previous observations substantial number of ILC2 and ILC3 subsets were present in the absence of IL-7R in the small intestinal lamina propria (siLP). Among ILC3 subsets, CCR6+ ILC3 population was most abundant in IL-7R deficient mice and expressed a high level of Bcl-2. ILC2 in the colon and lung were severely reduced in IL-7R deficient mice compared to those in siLP. The authors showed that residual ILC2 in siLP do not express ST2, a receptor for IL-33, and were less capable of producing IL-5 and IL-13 even in response to PMA and ionomycin. In contrast, all ILC3 subsets in IL-7R deficient mice were able to produce IL-22 as wild type ILC3s and provide protection against *C. rodentium* infection to some extent. The authors further showed that all ILC2 subsets as well as NK cells were virtually absent in mice lacking both IL-7R and IL-15 and conclude that ILC2 and ILC3 can persist through IL-15 signaling in the absence of IL-7 signaling.

Although the authors propose the role of IL-15 in the maintenance of ILC2 and ILC3, there is no data for the expression of IL-15R on ILCs. The authors need to show the expression patterns of IL-15R among ILC subsets in both wild type and IL-7R deficient mice.

It is not clear whether IL-15 plays a role in the differentiation or maintenance of ILC2 and ILC3s in the absence of IL-7R. The authors should transfer IL-7R deficient ILCs into IL-15 deficient mice to examine their fate. In addition, it is important to examine if IL-7R deficient ILCs can survive only in the small intestine after transfer to test the authors' hypothesis that IL-15 is abundantly expressed in the small intestine that supports ILCs in the small intestine.

Number of NK cells and ILC2 were greatly reduced in the colon of IL-7R deficient mice as shown in Figure 1F while those in siLP were unaffected. Are there any differences between colon and siLP for the mechanisms for generating/maintaining NK cells and ILC1?

The authors should examine the number of ILCs in adipose tissues such as mesentery of IL-7R deficient and mice lacking for both IL-7R and IL-15.

Below are other comments.

1) Figure 1B, E and H and Figure 5B: frequencies of ILCs should be shown as % of live CD45+CD3-CD19- cells instead of % of live CD45+ cells.

2) Page 6, third line: ref 23 should be added to refs 28-31.

3) Page 10, the last sentence: it is not true to state that "Because ILC2s are very rare in non-siLPs" because ILC2s are abundant in adipose tissues.

Reviewer #2 (Remarks to the Author):

The authors deal with the role of the IL-7 receptor for differentiation or maintenance of ILC. As previously reported, NK cells and ILC1 were not affected by genetic deficiency of the IL-7Ra, whereas ILC2 and ILC3 were reduced but not absent from lungs and small intestinal or colonic lamina propria. In *Il7ra*^{-/-} mice, ILC2 showed reduced levels of ST2 (a component of the IL-33R) and produced less IL-5 and IL-13. Residual ILC3 in *Il7ra*^{-/-} mice showed normal IL-22 production on a per cell level. Compared to *Rag*^{-/-} *Il2rg*^{-/-} mice, *Rag*^{-/-} *Il7ra*^{-/-} mice showed intermediate susceptibility to *C. rodentium* infection when compared to *Rag*^{-/-} mice. The subset of CCR6+ ILC3

was least affected by IL-7Ra deficiency which correlated with increased expression of Bcl2. The authors considered that IL-15 may contribute to the maintenance of ILC2 and ILC3 in Il7ra^{-/-} mice. Indeed, combined deficiency of IL-7Ra and IL-15 led to further reduction of ILC2 and ILC3 numbers.

The manuscript constitutes a thorough analysis of representation and function of all known ILC subsets in tissues with high representation of ILC (small intestine, colon, lung). As the authors acknowledge, similar analyses have been performed by others but a comprehensive side-by-side analysis of all known ILC subsets in IL-7Ra-deficient mice was not available. In general, the experiments seem carefully performed and support the major conclusions. However, there are several experimental issues and overstatements that need to be addressed.

1. The authors analyze ILC maintenance in adult Il7ra^{-/-} and Il15^{-/-} mice. The study should be expanded to provide a more complete overview during ontogeny (fetal period and postnatal). Such data would increase the significance of the manuscript considering that similar analyses in adult mice have been performed previously. In addition, it is not addressed if IL-7R signaling affects development, differentiation and/or maintenance of ILC.

2. To obtain a more complete picture of the relevance of cytokines signaling via the cytokine common gamma chain, Il2rg^{-/-} mice should be analyzed in parallel for the data presented in Figure 1 and Figure 5. In Figure 5, analysis of Il15^{-/-} mice should be added as well.

3. A thorough analysis of the expression (mRNA and protein) of gc cytokines should be performed for all tissues analyzed from Il7ra^{-/-}, Il15^{-/-} and control mice.

4. The authors analyze cytokine production by ILC2 after stimulation with IL-25 or a combination of IL-25, IL-33 and TSLP (Figure 2D,F). Is expression of IL-25R by ILC2 affected in the absence of IL-7Ra? It is not immediate obvious, what information stimulation of Il7ra^{-/-} ILC2 with IL-33 and TSLP will deliver, considering that both cytokines cannot signal because of reduced expression or absence of their cognate receptors.

5. In the discussion, the authors state (p. 11, line 237) that they performed a "comprehensive analysis of function by ST2⁺ and ST2⁻" ILC2. This is a bit of an overstatement as only 2 cytokines were analyzed after in vitro stimulation and physiological assays of ILC2 function were not performed.

6. The authors correlate the maintenance of ILC3 in Il7ra^{-/-} mice with increased expression of the antiapoptotic molecule Bcl2. However, this has not been experimentally tested. How are Bcl2 levels in the remaining ILC3 of Il7ra^{-/-} Il15^{-/-} DKO mice?

7. It is likely that microbial communities are affected in Il7ra^{-/-} mice and Il15^{-/-} mice. How was the impact of such changes discriminated from direct effects of IL-7R and IL-15 signaling?

8. The title reads like a bit of an overstatement because ILC2 and ILC3 are clearly reduced and functionally affected in Il7ra^{-/-} mice.

Reviewer #3 (Remarks to the Author):

This is an interesting paper describing the ILC populations present in gut of IL-7R KO mice. The data are of high quality and the conclusions generally convincing. However, there are some relevant missing information that are necessary to have a better understanding of the role of IL-7 and IL-15 in development and maintenance of ILCs in the gut.

I. It is interesting in Fig. 1 that there is a slight difference in the composition of ILC subsets in IL-7R KO mice between the siLP and colonic LP (cLP). Hence, NK cells were present in normal numbers in siLP, but sparse in cLP and ILC2 were more drastically reduced in cLP than siLP. These findings suggest a difference in the dependency for IL-7 among ILC subsets between siLP and cLP. Alternatively, it could be due to an indirect effect from the differential environmental changes in the small vs. large intestine from depletion of IL-7R-expressing cells. To gain an insight into these possibilities the authors should show expression levels of IL-7R on ILC subsets from siLP and cLP from WT mice compared to ILC subsets from the spleen.

II. While data from adult IL-7R KO mice are informative, there is a concern that some of the findings may not be physiologically relevant to WT mice due to the prolonged altered environment of the adult KO mice. This concern is especially relevant in the gut that harbor activated and memory lymphocytes previously been exposed to the food, commensal microbiota and local cytokine milieu. Hence, it is important to determine whether the main findings in the adult KO mice, such as ILC subsets and their effector function, also apply to neonatal IL-7R KO mice. This will provide an insight into whether the findings in IL-7R KO mice are due to the lack of IL-7 signaling or from the altered conditions of the intestine or both.

III. The three ILC2 subsets were not clearly shown, as Fig. 2B appears is on all ILCs. Hence, one cannot see the emergence of KLRG+ ST-2- population in IL-7R KO mice that is not detectable in WT mice. This should be rectified.

IV. The expression level of CD122 on the various subsets of ILCs in WT and IL-7R KO mice should be shown. This information could provide an insight into preferential survival of some of the ILC subsets in IL-7R KO mice. It is also possible that ILCs over-express CD122 as a mechanism to compensate for lack of IL-7R.

Reviewers' comments:

Reviewer #1 (Remarks to the Author):

Robinette et al. examined the role of IL-7R signaling in ILC differentiation and reported in this paper that unlike previous observations substantial number of ILC2 and ILC3 subsets were present in the absence of IL-7R in the small intestinal lamina propria (siLP). Among ILC3 subsets, CCR6+ ILC3 population was most abundant in IL-7R deficient mice and expressed a high level of Bcl-2. ILC2 in the colon and lung were severely reduced in IL-7R deficient mice compared to those in siLP. The authors showed that residual ILC2 in siLP do not express ST2, a receptor for IL-33, and were less capable of producing IL-5 and IL-13 even in response to PMA and ionomycin. In contrast, all ILC3 subsets in IL-7R deficient mice were able to produce IL-22 as wild type ILC3s and provide protection against C. rodentium infection to some extent. The authors further showed that all ILC2 subsets as well as NK cells were virtually absent in mice lacking both IL-7R and IL-15 and conclude that ILC2 and ILC3 can persist through IL-15 signaling in the absence of IL-7 signaling.

Although the authors propose the role of IL-15 in the maintenance of ILC2 and ILC3, there is no data for the expression of IL-15R on ILCs. The authors need to show the expression patterns of IL-15R among ILC subsets in both wild type and IL-7R deficient mice.

We thank the Reviewer for this excellent point. We have performed this experiment and find that all ILCs express CD122 in both WT and *IL7ra*^{-/-} mice. We also find that there is increased expression of CD122 by siLP ILCs in *IL7ra*^{-/-} mice, but not in the colon LP. We did not perform CD122 staining for lung ILC2 and adipose tissue ILC2 given the extremely low number of cells in the gate for these populations in *IL7ra*^{-/-} mice (Figure 1G and 1J). **These data has been added to a new Figures 6A-B.**

It is not clear whether IL-15 plays a role in the differentiation or maintenance of ILC2 and ILC3s in the absence of IL-7R. The authors should transfer IL-7R deficient ILCs into IL-15 deficient mice to examine their fate. In addition, it is important to examine if IL-7R deficient ILCs can survive only in the small intestine after transfer to test the authors' hypothesis that IL-15 is abundantly expressed in the small intestine that supports ILCs in the small intestine.

The proposed experiment would be a very elegant way to test our hypothesis that IL-15 sustains IL-7R deficient ILCs in the siLP. However, this experiment would be very technically challenging, given the very low total number of ILCs in *IL7ra*^{-/-} mice. In a preliminary feasibility experiment, we transferred a low number of WT DN ILC3s (20,000) to NSG mice and attempted to find them in the siLP 3 weeks later. Although we were able to detect ILC3s, they were mostly CCR6⁺ rather than DN, consistent with our findings reported here that CCR6⁺ ILC3s are relatively preserved in SCID mice (see new Figure 7A-B). Furthermore, ILCs did not differ between mice that received DN ILC3

transfers and those that did not. Although we cannot completely rule out the possibility that some transferred ILCs persisted among ILC3s, they were impossible to identify given the absence of congenic markers. Therefore, we expect this experiment would be very challenging if not impossible using lymphoreplete *Il15*^{-/-} mice, which similarly lack congenic markers.

Although we were unable to perform the requested experiment for the above reasons, we appreciate the Reviewer's point that we had not previously directed tested whether IL-15 could support ILC maintenance. There is some evidence in the literature that IL-15 can sustain ILCs, as mouse ILC3 have previously been cultured in IL-7 (20 ng/mL) and IL-15 (50 ng/mL) by Vonarbourg et al. To expand these prior results and to comprehensively test for the first time if IL-7 and IL-15 have differing capacities to sustain ILCs, we sorted ILC1/NK cells, CCR6⁺ ILC3, CCR6⁻ ILC3 (containing the T-bet^{lo/+} lineage of DN and NKp46⁺ ILC3s), and ILC2 from WT and *Il7ra*^{-/-} mice, and cultured equal numbers of cells from each genotype for 2 days in IL-7 (20 ng/mL), IL-15 (20 ng/mL and 50 ng/mL), or medium alone. We then assessed survival using 7aad/Annexin 5. We found that IL-15 significantly increased the survival of ILCs from both WT and *Il7ra*^{-/-} mice, while IL-7 only supported the survival of WT ILCs. We detected no difference in the ability of IL-15 to sustain ILC survival between genotypes, with the exception of CCR6⁺ ILC3, from which *Il7ra*^{-/-} cells survived significantly better in low dose IL-15 than WT cells. In general, CCR6⁺ ILC3s were resistant to γ_c cytokine depletion, and exhibited high survival in medium, with no significant differences in either genotype between any of our culture conditions. These data are consistent with our data demonstrating their high levels of Bcl-2 expression by CCR6⁺ ILC3s (Figure 5B-D; Supplemental 2C); their relative persistence in γ_c -family immunodeficient mice including *Il7ra*^{-/-}, *Il7ra*^{-/-}*Il15*^{-/-}, and *Rag2*^{-/-}*Il2rg*^{-/-} mice (Figure 1A-C, 7A-B); and prior data in the literature from Chappaz et al. showing similar γ_c -cytokine independent survival *in vitro*. **These data have been added to a new Figure 6C-J.** Notably, in Figure 6, we only show the percentage of cells that are 7aad⁺. While 7aad⁻ NK cells and ILC2 were also Annexin 5⁻, ILC3 auto fluorescence after culture prevented us from assessing this marker, per FMO controls from cultured ILC3s.

Number of NK cells and ILC2 were greatly reduced in the colon of IL-7R deficient mice as shown in Figure 1F while those in siLP were unaffected. Are there any differences between colon and siLP for the mechanisms for generating/maintaining NK cells and ILC1?

From the published literature, it is clear that a larger proportion of ILC1 in the colon than the siLP of WT mice are derived from ILC3 compared to those de novo generated in the bone marrow, based on the analysis of ROR γ t fate-mapping mice. However, converted ILC3s become indistinguishable from ILC1 in the absence of fate mapping and we are thus unable to assess the contribution of such plasticity here. Differences in mechanisms between NK cell generation in the colon and siLP are less clear. **In Figure 1N, we now demonstrate that there is an increased percentage of WT NK cells that express CD127 in the colon compared to the siLP and spleen, which we gated using FMO controls represented in Figure 1M.** Thus, it is possible that IL-7 has a greater effect on NK cells in the colon than in the siLP or spleen.

While our data for ILC2 and ILC3 subsets are very consistent throughout the paper, we find more heterogeneity in NK cell and ILC1 data between our experiments. Data for ILC1 and NK cells both appear visually different and were statistically significant between *IL7ra*^{-/-} and WT mice in the colon in Figure 1F, but were not significant in 7D. In Figure 4E, we demonstrated that the percentage of Ki67⁺ cells are significantly different between *IL7ra*^{-/-} and WT mice only for NK cells and ILC1. As experiments from each graph took place several months apart, it is possible that non-genetic differences such as those from the microbiota contribute to differences in the level of homeostatic proliferation we see between experiments. To this end, we have now tested the frequency and number of ILCs in *IL7ra*^{-/-} and WT mice on the day of birth, a time when all ILCs can be easily detected but there are minimal microbial effects. **These data are found in a new Figure 2.** Here, we observed a significant reduction in the frequency of colon NK cells between neonatal genotypes. There was also a trend toward a decreased number of NK cells in *IL7ra*^{-/-} colons versus WT colons that we did not observe in the siLP, but these data were not statistically significant. **We now highlight these ILC1 and NK cell results in our Results section and discuss this heterogeneity in the first paragraph of our Discussion.**

The authors should examine the number of ILCs in adipose tissues such as mesentery of IL-7R deficient and mice lacking for both IL-7R and IL-15.

This is a very good point that we had not previously included. We have performed these experiments in the inguinal fat of WT and *IL7ra*^{-/-} mice (**new graphs in Figures 1 K and L**), as well as in WT, *IL7ra*^{-/-}, *Il15*^{-/-}, *IL7ra*^{-/-}*Il15*^{-/-}, and *Rag2*^{-/-}*Il2rg*^{-/-} mice (**new graphs in Figures 7 F and G**). Similar to the lung and colon, we find that ILC2 are almost completely absent in gonadal adipose tissue in *IL7ra*^{-/-} mice, with no difference between *IL7ra*^{-/-}, and *IL7ra*^{-/-}*Il15*^{-/-}, and *Rag2*^{-/-}*Il2rg*^{-/-} mice in frequency or number.

Below are other comments.

1) Figure 1B, E and H and Figure 5B: frequencies of ILCs should be shown as % of live CD45+CD3-CD19- cells instead of % of live CD45+ cells.

This has been corrected.

2) Page 6, third line: ref 23 should be added to refs 28-31.

This has been corrected.

3) Page 10, the last sentence: it is not true to state that "Because ILC2s are very rare in non-siLPs" because ILC2s are abundant in adipose tissues.

This is an excellent catch of a poorly worded sentence. What we meant is that ILCs are very rare in *IL7ra*^{-/-} organs. **We have added the word "*IL7ra*^{-/-}" to clarify our meaning.**

Reviewer #2 (Remarks to the Author):

*The authors deal with the role of the IL-7 receptor for differentiation or maintenance of ILC. As previously reported, NK cells and ILC1 were not affected by genetic deficiency of the IL-7Ra, whereas ILC2 and ILC3 were reduced but not absent from lungs and small intestinal or colonic lamina propria. In *IL7ra*^{-/-} mice, ILC2 showed reduced levels of ST2 (a component of the IL-33R) and produced less IL-5 and IL-13. Residual ILC3 in *IL7ra*^{-/-} mice showed normal IL-22 production on a per cell level. Compared to *Rag*^{-/-} *Il2rg*^{-/-} mice, *Rag*^{-/-} *IL7ra*^{-/-} mice showed intermediate susceptibility to *C. rodentium* infection when compared to *Rag*^{-/-} mice. The subset of CCR6+ ILC3 was least affected by IL-7Ra deficiency which correlated with increased expression of Bcl2. The authors considered that IL-15 may contribute to the maintenance of ILC2 and ILC3 in *IL7ra*^{-/-} mice. Indeed, combined deficiency of IL-7Ra and IL-15 led to further reduction of ILC2 and ILC3 numbers.*

The manuscript constitutes a thorough analysis of representation and function of all known ILC subsets in tissues with high representation of ILC (small intestine, colon, lung). As the authors acknowledge, similar analyses have been performed by others but a comprehensive side-by-side analysis of all known ILC subsets in IL-7Ra-deficient mice was not available. In general, the experiments seem carefully performed and support the major conclusions. However, there are several experimental issues and overstatements that need to be addressed.

*1. The authors analyze ILC maintenance in adult *IL7ra*^{-/-} and *Il15*^{-/-} mice. The study should be expanded to provide a more complete overview during ontogeny (fetal period and postnatal). Such data would increase the significance of the manuscript considering that similar analyses in adult mice have been performed previously. In addition, it is not addressed if IL-7R signaling affects development, differentiation and/or maintenance of ILC.*

We appreciate these excellent points. We have now assessed ILC frequency and number in neonatal *IL7ra*^{-/-} siLP and colon LP. Here, we substantially reproduce our data from adult mice, demonstrating that siLP ILC2 and ILC3 develop in the absence of the IL-7R, with reduced total numbers. **These data have been added to a new figure, Figure 2.** We did not assess earlier fetal time points because

fetal ILCs are predominantly LT_i and other fetal ILCs, such as ILC2, are difficult to assess in the absence of the Arginase 1 reporter published by Bando et al.

The point that we had not previously established the mechanism by which IL-15 sustains IL-7R deficiency is also well taken. To this end, we now show that all ILCs express CD122. Additionally, CD122 expression is higher in *IL7ra*^{-/-} ILCs than WT ILCs. We also directly compare the effects of IL-15 versus IL-7 in the survival of WT and *IL7ra*^{-/-} ILCs in vitro. We demonstrate that all ILCs from both genotypes survive in IL-15. **These data have been added to a new Figure 6.**

IL-7 has been shown by Xu et al. regulate ILC progenitors by directly inducing Nfil3. However, in our *IL7ra*^{-/-}*Il15*^{-/-} mice, we previously provided no evidence if there were any additional changes to progenitor compartments when IL-15 was also removed. As we originally noted, identification of CHILP and ILCP was impossible due to the absence of CD127. However, we now evaluate ILC2P and rNKP and show that there are not greater reductions to these progenitors between single γ_c deficient animals, *IL7ra*^{-/-} vs *IL7ra*^{-/-}*Il15*^{-/-} for ILC2P and *Il15*^{-/-} vs *IL7ra*^{-/-}*Il15*^{-/-} for rNKP. **These data have been added to a new Figure 7H-K.**

2. To obtain a more complete picture of the relevance of cytokines signaling via the cytokine common gamma chain, Il2rg^{-/-} mice should be analyzed in parallel for the data presented in Figure 1 and Figure 5. In Figure 5, analysis of *Il15*^{-/-} mice should be added as well.

These are very good controls that we had not previously included in our analysis. Given that the experiments in Figure 1 and Figure 5 assessed the same parameters for siLP, we chose to retain the original content of Figure 1, but to repeat the content of Figure 5 with the *Il15*^{-/-} and *Rag2*^{-/-}*Il2rg*^{-/-} controls, **which are now found in a new Figure 7.** We further extended our original analysis from the figure to include all the other organs from Figure 1 (colon, lung, and adipose tissue). As mentioned above, we also include restricted bone marrow progenitors in the same conditions. We acknowledge that *Rag2*^{-/-}*Il2rg*^{-/-} are not the perfect control for this experiment, because Rag deficiency may increase restricted ILC progenitor frequencies on a T- and B-immunodeficient background or deleteriously effect ILCs as has previously been shown by the Karo et al. for NK cells. Yet, *Rag2*^{-/-}*Il2rg*^{-/-} are currently the most commonly used “ILC negative” control, and thus may pragmatically be the most useful control to the field for our study. Notably, we do not detect any significant differences between *IL7ra*^{-/-}*Il15*^{-/-} and *Rag2*^{-/-}*Il2rg*^{-/-} in any of our analyses, including among the restricted progenitor populations that we were able to assess in the absence of CD127.

*3. A thorough analysis of the expression (mRNA and protein) of γ_c cytokines should be performed for all tissues analyzed from *IL7ra*^{-/-}, *Il15*^{-/-} and control mice.*

We have now assessed the mRNA of all γ_c cytokines in small intestine, colon, and lung of the requested mice. We now show that that IL-15 is highly expressed compared to other γ_c cytokines, particularly in the GI tract. In addition, we detected no significant difference in mRNA expression between genotypes in any organ, with the exception of *Il15*, which was significantly reduced in *Il15*^{-/-} mice (data not shown). **This data can now be found in a new figure, Supplemental Figure 1.**

We did not assess γ_c cytokine protein levels given that the cytokines that most interest us, IL-7 and IL-15, are notoriously difficult to detect in tissues. Furthermore, even if we were able to detect protein levels, it would have been very challenging to interpret these data given that it would be unclear: 1- if cytokines were in the lamina propria interacting with ILCs or in other microenvironments such as the intraepithelium; 2- if IL-15 was appropriately transpresented; and 3- the impact of putative changes to cytokine levels given the concurrent changes to cytokine-consuming lymphocyte populations.

4. The authors analyze cytokine production by ILC2 after stimulation with IL-25 or a combination of IL-25, IL-33 and TSLP (Figure 2D,F). Is expression of IL-25R by ILC2 affected in the absence of IL-7Ra? It is not immediate obvious, what information stimulation of IL7ra^{-/-} ILC2 with IL-33 and TSLP will deliver, considering that both cytokines cannot signal because of reduced expression or absence of their cognate receptors.

These are helpful comments. We had not previously assessed IL-25R expression and now show that all siLP ILC2 subsets equally express this key cytokine receptor. We also show that there is no difference in expression between *IL7ra*^{-/-} and WT mice from the same experiment (**new graph, Figure 3D**).

The purpose of stimulating all ILC2s with IL-33 and TSLP was to compare *IL7ra*^{-/-} ILC2, which we expected to have diminished response to IL-33 and no response to TSLP, to fully competent WT ILC2s. Although we could have assessed IL-25 and PMA + ionomycin alone, this would not have shown if there was a residual capacity unique to WT cells engendered by their responsiveness to IL-33 and TSLP, well-recognized drivers of ILC2 cytokine production. **We have clarified this point in the text.**

5. In the discussion, the authors state (p. 11, line 237) that they performed a "comprehensive analysis of function by ST2+ and ST2-" ILC2. This is a bit of an overstatement as only 2 cytokines were analyzed after in vitro stimulation and physiological assays of ILC2 function were not performed.

This is a good point. **We have toned down our assessment of our findings to more fairly reflect**

the extent of our ILC2 functional analysis.

6. *The authors correlate the maintenance of ILC3 in IL7ra^{-/-} mice with increased expression of the antiapoptotic molecule Bcl2. However, this has not been experimentally tested. How are Bcl2 levels in the remaining ILC3 of IL7ra^{-/-} Il15^{-/-} DKO mice?*

Another excellent suggestion. We have now concurrently evaluated Bcl-2 expression between our extended set of mice that this Reviewer previously suggested, namely WT, *IL7ra^{-/-}*, *Il15^{-/-}*, *IL7ra^{-/-} Il15^{-/-}*, and *Rag2^{-/-} Il2rg^{-/-}* mice. In this extended analysis, we find that DN ILC3 Bcl-2 upregulation was completely γ_c -dependent, but IL-15-independent; CCR6⁺ ILC3 Bcl-2 upregulation was partially γ_c -independent but was also fully IL-15-independent. Therefore, although DN and CCR6⁺ ILC3 frequency correlated with their increased Bcl-2 expression in the periphery of *Il7ra^{-/-}* mice, this mechanism did not explain the role of IL-15 in ILC3 maintenance. **We have added these data to Supplemental Figure 2C and discuss them in paragraph 7 of the Discussion.**

7. *It is likely that microbial communities are affected in IL7ra^{-/-} mice and Il15^{-/-} mice. How was the impact of such changes discriminated from direct effects of IL-7R and IL-15 signaling?*

Though beyond the scope of this paper, it is definitely possible and perhaps even likely that 1- differences in microbial communities and/or infection may differently effect ILC generation in the presence or absence of IL-7 or IL-15, reminiscent of data previously shown by Sun et al. for MCMV-induced NK cell generation in γ_c deficient mice; 2- immunodeficiency may differently affect the microbial communities contained in the mice, reminiscent of the data of Sonnenberg et al. that demonstrated microbial differences in ILC-depleted mice; and 3- microbial changes may be progressive over generations of mice, similar to those induced by diet through generations as recently shown by Sonnenburg et al. Indeed, such questions may be particularly interesting and relevant to patients and the children of patients with non-lethal primary immunodeficiency, such as selective IgA immunodeficiency or CVID, and are worthy of further investigation.

For our study, we maintained all mice in the same SPF facility, with the exception of *Rag2^{-/-} Il2rg^{-/-}* mice, which were purchased from Taconic and were cohoused with our mice prior to use. Furthermore, as discussed above, we have now assessed ILC frequency and numbers in *IL7ra^{-/-}* and WT mice on the day of birth, a time when all ILCs can be easily detected but there are minimal microbial effects, **in Figure 2**. Here, we reproduce our data from adult mice, suggesting that the effects we measure in our mice are not purely a result of differences in the microbiota. **However, we also now acknowledge a limited degree of ILC1 and NK cell variability between similar experiments in adult mice that may represent differences in homeostatic proliferation secondary to non-genetic causes such as the microbiota.**

8. *The title reads like a bit of an overstatement because ILC2 and ILC3 are clearly reduced and functionally affected in IL7ra^{-/-} mice.*

We thank the Reviewer for bringing this important issue to our attention. Our title was meant to be provocative, but we definitely do not intend to mislead our readers about the content of our data. **We will discuss this point further with the editor and are of course open to any suggestions from any Reviewers regarding a superior title.**

Reviewer #3 (Remarks to the Author):

This is an interesting paper describing the ILC populations present in gut of IL-7R KO mice. The data are of high quality and the conclusions generally convincing. However, there are some relevant missing information that are necessary to have a better understanding of the role of IL-7 and IL-15 in development and maintenance of ILCs in the gut.

I. It is interesting in Fig. 1 that there is a slight difference in the composition of ILC subsets in IL-7R KO mice between the siLP and colonic LP (cLP). Hence, NK cells were present in normal numbers in siLP, but sparse in cLP and ILC2 were more drastically reduced in cLP than siLP. These findings suggest a difference in the dependency for IL-7 among ILC subsets between siLP and cLP. Alternatively, it could be due to an indirect effect from the differential environmental changes in the small vs. large intestine from depletion of IL-7R-expressing cells. To gain an insight into these possibilities the authors should show expression levels of IL-7R on ILC subsets from siLP and cLP from WT mice compared to ILC subsets from the spleen.

This is an excellent observation from our data. We have performed the suggested experiment and now show CD127 expression in WT mice across all ILCs we analyze in this paper (**new Figure 1M**). Quantifying these results, we find that a greater frequency of colon NK cells express CD127 than siLP and spleen NK cells; siLP NK cells also expressed CD127 more than spleen NK cells (**new Figure 1N**). **We also now discuss these data in the first paragraph of our Discussion.**

II. While data from adult IL-7R KO mice are informative, there is a concern that some of the findings may not be physiologically relevant to WT mice due to the prolonged altered environment of the adult KO mice. This concern is especially relevant in the gut that harbor activated and memory lymphocytes previously been exposed to the food, commensal microbiota and local cytokine milieu. Hence, it is important to determine whether the main findings in the adult KO mice, such as ILC subsets and their effector function, also apply to neonatal IL-7R KO mice. This will provide an insight

into whether the findings in IL-7R KO mice are due to the lack of IL-7 signaling or from the altered conditions of the intestine or both.

We appreciate these excellent points. We have now assessed ILC frequency and number in neonatal *IL7ra*^{-/-} siLP and colon LP. Here, we substantially reproduce our data from adult mice, demonstrating that siLP ILC2 and ILC3 develop in the absence of the IL-7R, with reduced total numbers. **These data have been added to a new figure, Figure 2.** Unfortunately, we were unable to evaluate cytokine production by neonatal ILCs because of technical challenges created by the very small total number lymphocytes in neonatal mice.

III. The three ILC2 subsets were not clearly shown, as Fig. 2B appears is on all ILCs. Hence, one cannot see the emergence of KLRG+ ST-2- population in IL-7R KO mice that is not detectable in WT mice. This should be rectified.

This is a good catch. **We mistakenly failed to label the first plot as also gated on “GATA3^{hi}” ILC2s, which has now been corrected.** Thus, the first plot in Figure 2B (now Figure 3A) actually shows the phenotype of ILC2s. The next plot in the graph was a control, showing our gating for the CCR6⁺ ILC3s, which serve as the RORγt positive control. **We agree with the Reviewer that this second plot was superfluous and confusing so it has been removed.** Finally, the point that there are 3 ILC2 subsets rather than the 2 we previously discuss is also well taken. **We now provide quantification for all three subsets with the KLRG1⁻ST2⁻ ILC2s and show that they are similarly RORγt⁻, but IL25R⁺ (Figure 3B, D-E).**

IV. The expression level of CD122 on the various subsets of ILCs in WT and IL-7R KO mice should be shown. This information could provide an insight into preferential survival of some of the ILC subsets in IL-7R KO mice. It is also possible that ILCs over-express CD122 as a mechanism to compensate for lack of IL-7R.

We thank the Reviewer for this excellent point. We have performed this experiment and find that all ILCs express CD122 in both WT and *IL7ra*^{-/-} mice. We also find that there is increased expression of CD122 by siLP ILCs in *IL7ra*^{-/-} mice, but not in the colon LP. We did not perform CD122 staining for lung ILC2 and adipose tissue ILC2 given the extremely low number of cells in the gate for these populations in *IL7ra*^{-/-} mice (Figure 1G and 1J). **These data has been added to a new Figure 5A-B.**

We now also directly test the role of IL-7 and IL-15 in the maintenance of siLP ILCs through a new set of in vitro experiments. To compare IL-7 and IL-15 effects directly, we sorted ILC1/NK

cells, CCR6⁺ ILC3, CCR6⁻ ILC3 (containing the T-bet^{lo/+} lineage of DN and NKp46⁺ ILC3s), and ILC2 from WT and *IL7ra*^{-/-} mice, and cultured equal numbers of cells from each genotype in IL-7 (20 ng/mL), IL-15 (20 ng/mL and 50 ng/mL), or medium alone. After 2 days, we assessed ILC survival. We found that IL-15 significantly increased the survival of ILCs from both WT and *IL7ra*^{-/-} mice, while IL-7 only supported the survival of WT ILCs. Although CD122 expression was higher in *IL7ra*^{-/-} mice, we detected no difference in the ability of IL-15 to sustain ILC survival between genotypes, though *in vitro* conditions may imperfectly reflect differences in IL-15 signaling *in vivo*. The exception to this was CCR6⁺ ILC3, as *IL7ra*^{-/-} cells survived significantly better in low dose IL-15 than WT cells. In general, CCR6⁺ ILC3s were resistant to γ_c cytokine depletion, and exhibited high survival in medium, with no significant differences in either genotype between any of our culture conditions. These data are consistent with our data demonstrating their high levels of Bcl-2 expression by CCR6⁺ ILC3s (Figure 5B-D; Supplemental Figure 2C); their relative persistence in γ_c -family immunodeficient mice including *IL7ra*^{-/-}, *IL7ra*^{-/-}*IL15*^{-/-}, and *Rag2*^{-/-}*Il2rg*^{-/-} mice (Figure 1A-C, 7A-B); and prior data in the literature from Chappaz et al. showing similar γ_c -cytokine independent survival *in vitro*. **To our knowledge, these data are the first to directly compare the capacity of IL-7 and IL-15 to sustain ILCs, and have been added to the same new figure, 5C-F.**

REVIEWERS' COMMENTS:

Reviewer #1 (Remarks to the Author):

The authors have addressed most of the reviewers' comments and the paper has been improved substantially. There are, however, a few minor points that need to be clarified.

- 1) There are two panels for Figure 7D, which should be corrected.
- 2) On page 13, the authors stated IL7Ra deficient CCR6+ILC3 survived better in low dose of IL-15 than wild type cells by citing Figure 6F. However, it is not obvious from the figure
- 3) In the last line of page 8, "cells" and "cultured" were reversed.

Reviewer #2 (Remarks to the Author):

This is a revised version of this manuscript. The authors have thoroughly addressed my points and the manuscript seems much improved.

Reviewer #3 (Remarks to the Author):

The revised version of the paper has addressed all the concerns.

We appreciate the final comments and enthusiasm from the Reviewers. Please see our responses below.

Reviewer #1 (Remarks to the Author):

The authors have addressed most of the reviewers' comments and the paper has been improved substantially. There are, however, a few minor points that need to be clarified.

1) There are two panels for Figure 7D, which should be corrected.

This has been corrected.

2) On page 13, the authors stated IL7Ra deficient CCR6+ILC3 survived better in low dose of IL-15 than wild type cells by citing Figure 6F. However, it is not obvious from the figure

We did not change our wording on page 13 due to word constraints. There is a blue asterisk in Figure 7 panel F that indicates statistical significance. As indicated in the figure legend, "Colored asterisks between graphs correspond to significant differences between genotypes, color-coded to the key to the right of each graph." We agree with the referee that the difference appears minimal but acknowledge it in the text because it did reach statistical significance.

3) In the last line of page 8, "cells" and "cultured" were reversed.

This has been corrected.

Reviewer #2 (Remarks to the Author):

This is a revised version of this manuscript. The authors have thoroughly addressed my points and the manuscript seems much improved.

We are happy that we have improved our manuscript and "thoroughly" addressed critiques.

Reviewer #3 (Remarks to the Author):

The revised version of the paper has addressed all the concerns.

We are happy that we have addressed "all the concerns."